# Tradeoffs in alignment and assembly-based methods for structural variant detection with long-read sequencing data

Yichen Henry Liu[1,4], Can Luo[2,4], Staunton G. Golding[2], Jacob B. Ioffe[1] & Xin Maizie Zhou [1,2,3] ✉

Long-read sequencing offers long contiguous DNA fragments, facilitating diploid genome assembly and structural variant (SV) detection. Efficient and robust algorithms for SV identification are crucial with increasing data availability. Alignment-based methods, favored for their computational efficiency and lower coverage requirements, are prominent. Alternative approaches, relying solely on available reads for de novo genome assembly and employing assembly-based tools for SV detection via comparison to a reference genome, demand significantly more computational resources. However, the lack of comprehensive benchmarking constrains our comprehension and hampers further algorithm development. Here we systematically compare 14 read alignment-based SV calling methods (including 4 deep learning-based methods and 1 hybrid method), and 4 assembly-based SV calling methods, alongside 4 upstream aligners and 7 assemblers. Assembly-based tools excel in detecting large SVs, especially insertions, and exhibit robustness to evaluation parameter changes and coverage fluctuations. Conversely, alignment-based tools demonstrate superior genotyping accuracy at low sequencing coverage (5-10×) and excel in detecting complex SVs, like translocations, inversions, and duplications. Our evaluation provides performance insights, highlighting the absence of a universally superior tool. We furnish guidelines across 31 criteria combinations, aiding users in selecting the most suitable tools for diverse scenarios and offering directions for further method development.

Human genome variations include single nucleotide variations (SNVs), small insertions and deletions (indels), and structural variants (SVs). SNVs and small indels are defined as alterations of the genome smaller than 50 base pairs, and dominate genomic variations in terms of absolute number. SVs are larger alterations that span >50 base pairs, including deletions (DELs), duplications (DUPs), inversions (INVs), insertions (INSs), and translocations (TRAs)[1], which contribute far more to sequence divergence given their larger size[2]. Thus, SVs can make up the majority of an organism's variation[3]. SVs have been

implicated in many diseases and pathogenic conditions, including increasing risk for brain disorders such as Parkinson's and Alzheimer's diseases, immune system complications, and organ malformations[4–7].

To fully explore the role of SVs in disease conditions and in individual genomic variability, short-read and long-read whole genome sequencing have been used along with associated analytical methods to characterize SVs. Short-read sequencing, also known as next-generation sequencing technology, has been developed during the past decades[8]. Their high throughput and high sequencing accuracy

[1]Department of Computer Science, Vanderbilt University, 37235 Nashville TN, USA. [2]Department of Biomedical Engineering, Vanderbilt University, 37235 Nashville TN, USA. [3]Data Science Institute, Vanderbilt University, 37235 Nashville TN, USA. [4]These authors contributed equally: Yichen Henry Liu, Can Luo. ✉ e-mail: maizie.zhou@vanderbilt.edu

paved the way for high performance in SNV and small indel detection. However, due to their short length and lack of context within the genome, it is much more challenging for short reads to accurately detect and cover the full length of SVs, particularly within repetitive regions, and more so for larger SVs[9].

The advent of long-read sequencing technologies, mainly represented by Pacific Biosciences (PacBio)[10] and Oxford Nanopore Technologies (ONT)[11], addressed many of the limitations of short reads. Long-read sequencing technologies typically generate reads with over 10 kilobases average length, and thus lead to higher confidence arrangement and accuracy[12]. Pacbio single-molecule, real-time (SMRT) sequencing, produces either Continuous Long Reads (CLRs) or High-Fidelity (HiFi) reads. Pacbio CLR can be generated from RS II, Sequel, and Sequel II sequencing platforms with ~85% read accuracy, and can reach a maximum 200 kb read length on the Sequel II platform. Pacbio Hifi reads generated via circular consensus sequencing (CCS) can reach 99.9% accuracy on the Sequel II platform. The latest Pacbio Revio platform achieves exceptional performance, with median read accuracy reaching 99.9% and with a 15-time increase in HiFi read throughput compared to Sequel II[13]. Pacbio also recently introduced the Onso platform that generates highly accurate short reads with PacBio sequencing by binding (SBB) technology[14]. ONT sequencing, on the other hand, is able to generate reads with over 1000 kb maximum read length and 87–98% accuracy[15]. While early long-read sequencing was hampered by lower throughput, higher error rates, and higher cost per base[16], the latest techniques can now generate long reads with high accuracy (such as PacBio HiFi), throughput (such as ONT PromethION), and cost per base (also ONT PromethION) comparable to Illumina short-reads[15].

The discovery and accurate characterization of SVs are of primary interest when analyzing long reads, and this process requires specific long-read SV calling methods. Whole genome data from an individual are typically aligned to a reference genome (read alignment-based) to identify variants. Read alignment-based approaches are less computationally demanding and require less sequencing coverage. A plethora of read alignment-based SV callers has emerged in recent years, such as PBHoney[17], NanoSV[18], Smartie-sv[19], Sniffles[20], SVIM[21], cuteSV[22], NanoVar[23], pbsv[24], SKSV[25], Sniffles2[26], MAMnet[27] and DeBreak[28]. An alternative approach is to assemble the whole genome of an individual based on their reads alone (de novo assembly) and compare the assembly with a reference genome, which is much more demanding in computational resources. To date, only a few assembly-based SV callers have been introduced, such as Dipcall[29], Smartie-sv[19], SVIM-asm[30], and PAV[31]. The dichotomization of alignment and assembly-based methods is not absolute. Smartie-sv[19], has both read alignment-based and assembly-based modes. DeBreak, can also be classified as a hybrid method that employs a read alignment-based strategy in combination with a local assembly approach to detect SVs. NanoVar and MAMnet can be classified as deep learning-based approaches relying on read alignment information. For simplicity, we classified the hybrid or deep learning-based methods as read alignment-based since they rely on read alignment first in their framework to extract features.

In this work, we first systematically analyze and evaluate the performance of 16 state-of-the-art long-read SV methods, introduced between the years 2014 and 2022 on a multitude of PacBio and ONT datasets. We specifically investigate the relative strengths and differences of read alignment-based versus assembly-based SV methods. We additionally examine the effect of upstream tools such as aligners or assemblers, on their SV calling performance. Finally, we use our comprehensive evaluation framework to benchmark two more deep learning-based SV calling methods published most recently in late 2022 and 2023. Assembly-based tools exhibit a higher sensitivity in detecting large structural variants (SVs), especially insertions, and demonstrate greater robustness to changes in evaluation parameters and sequencing coverage. Conversely, alignment-based tools excel in

genotyping accuracy at low sequencing coverage (5–10×) and are adept at detecting complex SVs such as translocations, inversions, and duplications. However, assembly-based calling pipelines typically demand more computational time. Despite these trade-offs, no single tool achieves consistently high and robust performance across all conditions. To aid users in tool selection, we provide guidelines and recommendations encompassing 31 combinations of criteria. Our work provides guidance to stimulate future method and tool development that increases accuracy, robustness, efficiency, and reproducibility in SV discovery and evaluation.

## Results
We first investigated SV detection in 12 alignment-based methods (including deep learning-based methods NanoVar and MAMnet, and hybrid method DeBreak) and 4 assembly-based methods, in 11 PacBio Hifi, CLR, and ONT datasets, 9 simulated long reads datasets, and 2 paired tumor-normal CLR and ONT datasets. We then evaluated their performance in terms of downstream analyses (Fig. 1). One tool, Smartie-sv[19], has both read alignment-based and assembly-based modes, and we therefore treated it as two methods: "Smartie-sv_aln" and "Smartie-sv_asm" in the paper. In the last section, we additionally benchmarked two most recent deep learning-based SV calling methods (SVision[32] and INSnet[33]) using our comprehensive evaluation framework and discussed their strengths and weaknesses compared to traditional alignment-based methods. Among the 11 long-read sequencing datasets (Table 1), five PacBio HiFi datasets were referred to as Hifi_L1, Hifi_L2, Hifi_L3, Hifi_L4, and Hifi_L5. They had ~56.3×, 30×, 34×, 28×, and 41× coverage, respectively. Three PacBio CLR datasets were referred to as CLR_L1, CLR_L2, and CLR_L3, and their coverage was 65.1×, 88.6×, and 28.6×, respectively. We also used three ONT datasets referred to as Nano_L1, Nano_L2, and Nano_L3. Their coverage was ~45.6×, 57×, and 48×. We used Hifi_L1 and Nano_L1 as representative libraries to demonstrate some general statistics of SV calling performance. More information for each SV caller and dataset is provided in Tables 1, 2. We also designed a user recommendation table (Table 3 and Supplementary Table 1) along with the following results to highlight the best performance or robust tools.

### SV length distribution and calling performance under a set of moderate-tolerance parameters
We first examined the SV length distribution performance of all methods. Figure 2a–p shows the general trend of the length distribution of SVs from 12 read alignment-based and 4 assembly-based methods on Hifi_L1 and Nano_L1. Some tools were not applicable to ONT datasets. The number of SVs decreased sharply as a function of size, with most SVs clustering in the 50–400 bp range. We found that alignment-based calling tools had similar length distributions for deletions, while they varied considerably for insertions. Assembly-based tools showed a similar trend in deletion length distribution as read alignment-based tools, while detecting more large insertions than most alignment-based tools, especially for insertions over 1 kb in Hifi_L1. Compared to Hifi_L1, Nano_L1 had less coverage, which could affect the contiguity of assembled sequences and further impair the SV detection. To further explore the effect of different sequencing coverages on all SV callers, we performed multiple SV calling and evaluation experiments on subsampled datasets (see Subsampling effects on SV calling section, below). Additional details for the SV length distribution can be found in supplementary notes section 2.1.

SVs of different sizes could be detected with different accuracy rates by the tools we examined. Therefore, we plotted F1 score as a function of SV size range. We started the performance evaluation with a set of fixed parameters in Truvari[34] that were neither too stringent nor too relaxed ($p = 0$, $P = 0.5$, $r = 500$, $O = 0$); we refer to these as modest tolerance parameters (Fig. 2q–r). The parameter $p$ was set to

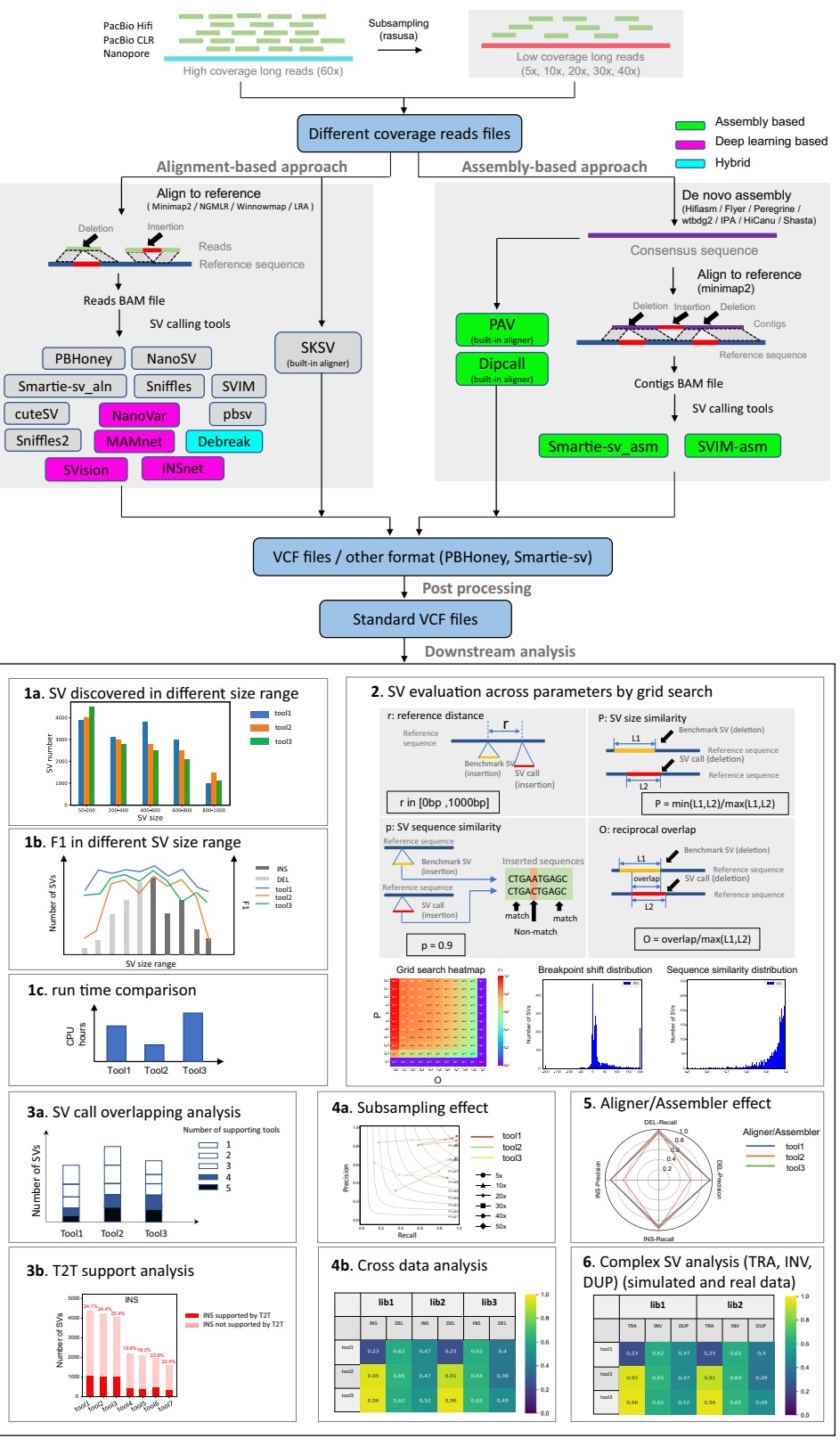

**Fig. 1 | The workflow for benchmarking alignment- and assembly-bassed long-read SV callers.** The inputs are PacBio Hifi, Pacbio CLR and ONT datasets and their subsampled datasets at different sequencing coverages. The whole benchmarking workflow includes 12 read alignment-based SV calling methods, 4 assembly-based SV calling methods, 4 aligners, and 7 assemblers. It involves several different downstream analyses to evaluate all these SV callers. Assembly-based methods are highlighted in green, deep learning-based methods are highlighted in purple, the hybrid method is highlighted in cyan, and traditional alignment-based methods are not highlighted.

**Table 1 | Benchmark datasets**

| Dataset | Abbreviation in the paper | Coverage | Source |
|---|---|---|---|
| HG002 Pacbio CCS 15kb+20 kb | Hifi_L1 | 56.3× | https://ftp-trace.ncbi.nlm.nih.gov/ReferenceSamples/giab/data/AshkenazimTrio/HG002_NA24385_son/PacBio_CCS_15kb_20 kb_chemistry2/reads/ |
| HG002 PacBio CCS 10 kb | Hifi_L2 | 30× | https://ftp-trace.ncbi.nlm.nih.gov/ReferenceSamples/giab/data/AshkenazimTrio/HG002_NA24385_son/PacBio_CCS_10 kb/ |
| HG002 PacBio CCS 11kb | Hifi_L3 | 34× | https://www.ncbi.nlm.nih.gov/sra/SRR8833180 |
| HG002 PacBio CCS 15kb | Hifi_L4 | 28× | https://ftp-trace.ncbi.nlm.nih.gov/ReferenceSamples/giab/data/AshkenazimTrio/HG002_NA24385_son/PacBio_CCS_15kb/ |
| HG002 PacBio CCS 16kb | Hifi_L5 | 41× | https://www.ncbi.nlm.nih.gov/bioproject/PRJNA832505 |
| HG002 MtSinai | CLR_L1 | 65.1× | https://ftp-trace.ncbi.nlm.nih.gov/ReferenceSamples/giab/data/AshkenazimTrio/HG002_NA24385_son/PacBio_MtSinai_NIST/ |
| HG002 Pacbio CLR | CLR_L2 | 88.6× | https://www.ncbi.nlm.nih.gov/sra/SRX7668835 |
| HG002 Pacbio CLR | CLR_L3 | 28.6× | https://www.ncbi.nlm.nih.gov/sra/SRX6719924 |
| HG002 Nanopore Promethion | Nano_L1 | 45.6× | https://ftp-trace.ncbi.nlm.nih.gov/ReferenceSamples/giab/data/AshkenazimTrio/HG002_NA24385_son/UCSC_Ultralong_OxfordNanopore_Promethion/ |
| HG002 Nanopore UL guppy3.2.4 | Nano_L2 | 57× | ftp://ftp-trace.ncbi.nlm.nih.gov/ReferenceSamples/giab/data/AshkenazimTrio/HG002_NA24385_son/Ultralong_OxfordNanopore/guppy-V3.2.4_2020-01-22/ |
| HG002 Nanopore PRJNA678534 | Nano_L3 | 48× | https://www.ncbi.nlm.nih.gov/Traces/study/?acc = SRP292617&o = acc_s%3Aa |
| 9 Simulated long reads datasets | Hifi_TRA Hifi_DUP Hifi_INV CLR_TRA CLR_DUP CLR_INV ONT_TRA ONT_DUP ONT_INV | 40× | CLR&ONT: VISOR+PBSIM3; Hifi: VISOR+PBSIM3+CCS |
| HCC1395 Tumor Pacbio | HCC1395_PB | 39× | https://www.ncbi.nlm.nih.gov/sra/?term = SRR8955953 |
| HCC1395 Normal Pacbio | HCC1395BL_PB | 44× | https://www.ncbi.nlm.nih.gov/sra/?term = SRR8955954 |
| HCC1395 Tumor ONT | HCC1395_ONT | 12× | https://www.ncbi.nlm.nih.gov/sra/?term = SRR16005301 |
| HCC1395 Normal ONT | HCC1395BL_ONT | 19× | https://www.ncbi.nlm.nih.gov/sra/?term = SRR17096031 |

For simulated datasets, the corresponding simulators are listed in the source field.

zero to disable SV sequence comparison since tools such as PBHoney, Smartie-sv, NanoVar, MAMnet, and DeBreak do not provide alternate allele sequences for insertions. The parameter $O$ was set to zero to allow breakpoint shift for deletions, which was favorable for most alignment-based tools. The parameter $P$ and $r$ used the mean value of the range of each parameter, respectively. A more detailed investigation of these parameters for SV validation is discussed in the following section. The results on Hifi_L1 indicated that all four assembly-based tools (Dipcall, Smartie-sv_asm, PAV, and SVIM-asm) and three of the alignment-based tools (SKSV, cuteSV, and MAMnet) were robust to changes of SV size. The performance of the rest alignment-based tools dropped markedly when the SV size was in different ranges. PBHoney and Smartie-sv_aln did not perform well for both deletions larger than 4kb and insertions larger than 1 kb. Sniffles, Sniffles2, SVIM, and NanoVar did not perform well for insertions larger than 1 kb. DeBreak, pbsv, NanoVar, and PBHoney had a performance drop for small insertions in the range of 50–200 bp. NanoSV had stable, but low F1 scores across different size ranges. In short, assembly-based tools were most robust to SV size changes. In Nano_L1 (Fig. 2r), performance was similar to what we observed in Hifi_L1 for the same tools, except for DeBreak, SVIM, and Sniffle2. They had a relatively more robust performance on small (50–200 bp) and large (>1 kb) insertions in Nano_L1, compared to Hifi_L1. The total number of SV calls for each tool on both Hifi_L1 and Nano_L1 is shown in Supplementary Table 2. We also show true positives, false positives, false negatives, total benchmark SV calls, recall, precision, F1, and genotyping accuracy for three different size ranges of all tools in Supplementary Tables 3–5 on both Hifi_L1 and Nano_L1. The repeat annotation analysis for SVs was further investigated in Supplementary Fig. 1. The results indicated that the SV accuracy of long-read SV callers was not affected by different types of repeats. More detailed results of repeat analysis were also shown in supplementary notes section 2.3.

## Computation cost

To investigate the runtime of 16 SV calling methods, 4 aligners, and 7 assemblers, we plotted and compared CPU hours. We classified tools into three tiers depending on CPU time, though these were somewhat dependent on the library. For Hifi_L1 (black bars in Fig. 2s–t and Supplementary Table 6), cuteSV, MAMnet, SVIM, PBHoney, Sniffles, Sniffles2, Dipcall, and Smartie-sv_aln finished within 3-17 CPU hours; SVIM-asm, pbsv, SKSV, and NanoVar finished within 36–64 CPU hours; PAV, Smartie-sv_asm, DeBreak, and NanoSV finished within 116-863 CPU hours. For Nano_L1 (gray bars in Fig. 2s–t and Supplementary Table 6), cuteSV, Sniffile, SVIM, SVIM-asm, and Dipcall finished within 5-24 CPU hours; NanoVar finished within 95 CPU hours; NanoSV, Smartie-sv_aln, and PAV finished within 204-2760 CPU hours. With respect to memory usage, SV callers often consumed <100 Gb memory, except for PAV and Smartie-sv_aln when processing Nano_L1 (Supplementary Table 6 and Supplementary Table 7).

We also evaluated compute time of different aligners and assemblers (Fig. 2u–v and Supplementary Table 7). Aligner compute times were generally modest, and order of performance depended on the library. CPU hours varied between 97 for NGMLR[20] to 593 for minimap2[35]. Longer compute times and greater variance in CPU time were observed for different assemblers. These varied between 440 for Hifiasm to 5386 for Peregrine. More details are provided in supplementary notes section 2.2.

## Evaluation of SV calls across parameters

Structural variants often span a large region, and benchmarking them is challenging compared to SNVs. When benchmarking SV calls of each tool against the GIAB gold standard SV callset, comparisons do not usually require an exact match that needs two SVs to be identical in terms of breakpoints and SV sequence. Instead, SVs are considered the same as long as their differences and similarities are under a set of pre-

## Table 2 | Benchmark tools

| SV callers | Version | Resource | Method | SV types | Link |
|---|---|---|---|---|---|
| PBHoney | 15.8.24 | English et al. 2014[17] | Alignment discordance signal | DEL, INS | http://deb.debian.org/debian/pool/main/p/pbsuite/ |
| NanoSV | 1.2.4 | Stancu et al. 2017[18] | Breakpoint junction clustering | DEL, INS, DUP, BND | https://github.com/mroosmalen/nanosv |
| Smartie-sv_aln | default | Kronenberg et al. 2018[19] | N/A | DEL, INS | https://github.com/zeeev/smartie-sv |
| Sniffles | 1.0.12 | Sedlazeck et al. 2018[20] | Signature clustering | DEL, INS, DUP, INV, TRA, INVDUP | https://github.com/fritzsedlazeck/Sniffles |
| SVIM | 1.4.2 | Heller et al. 2019[21] | Signature clustering | DEL, INS, DUP, INV, BND, DUP:TANDEM, DUP:INT | https://github.com/eldariont/svim |
| cuteSV | 1.0.11 | Jiang et al. 2020[22] | Signature clustering | DEL, INS, DUP, INV, BND | https://github.com/tjiangHIT/cuteSV |
| NanoVar | 1.3.9 | Tham 2020[23] | Novel adjacency + ANN (DL) | DEL, INS, DUP, INV, BND | https://github.com/benoukraflab/NanoVar |
| pbsv | 2.6.2 | | Signature clustering | DEL, INS, DUP, INV, BND, CNV | https://github.com/PacificBiosciences/pbsv |
| SKSV | 1.0.2 | Liu et al. 2021[25] | Signature clustering | DEL, INS, DUP, INV, BND | https://github.com/ydLiu-HIT/SKSV |
| Sniffles2 | 2.0.6 | Smolka et al. 2022[26] | Signature clustering | DEL, INS, DUP, INV, BND | https://github.com/fritzsedlazeck/Sniffles |
| MAMnet | default | Ding et al. 2022[27] | CNN+LSTM (DL) | DEL, INS | https://github.com/micahvista/MAMnet |
| DeBreak | 1.0.2 | Yu et al. 2023[28] | Signature clustering + local assembly (Hybrid) | DEL, INS, DUP, INV, TRA | https://github.com/Maggi-Chen/DeBreak |
| SVision | 1.3.8 | Lin et al. 2022[32] | CNN (DL) | DEL, INS, DUP, tDUP, INV | https://github.com/xjtu-omics/SVision |
| INSnet | default | Gao et al. 2023[33] | CNN+GRU (DL) | INS | https://github.com/eioyuou/INSnet |
| Dipcall | 0.3 | Li et al. 2018[29] | N/A | DEL, INS, SNV | https://github.com/lh3/dipcall |
| Smartie-sv_asm | default | Kronenberg et al. 2018[19] | N/A | DEL, INS | https://github.com/zeeev/smartie-sv |
| SVIM-asm | 1.0.2 | Heller et al. 2020[30] | Signature | DEL, INS, DUP, INV, BND, DUP:TANDEM, DUP:INT | https://github.com/eldariont/svim-asm |
| PAV | freeze2 | Ebert et al. 2021[31] | Refined breakpoints + k-mer density | DEL, INS, INV, SNV | https://github.com/EichlerLab/pav |

DL stands for deep learning.

defined thresholds. In the previous section for the evaluation of SV calls we selected a set of fixed parameters from Truvari to demonstrate the performance of all SV callers. However, the choice of these error tolerances is usually subjective, and just a single set of thresholds could limit our understanding of the performance of different SV callers. A relaxed threshold could underestimate the differences among SV callers, whereas a stringent threshold could overstate the advantage of a specific SV caller. Thus, to thoroughly and comprehensively reveal the characteristics and performance of these alignment- and assembly-based SV calling methods, we explored a set of grid searches on Truvari parameters including pctstim (*p*), pctsize (*P*), pctovl (*O*), and refdist (*r*) to investigate the robustness of each SV caller. Among these parameters, *p* controls the minimum allele sequence similarity used to identify two SV calls as the same; *P* corresponds to the minimum allele size similarity between the compared SVs; *O* determines the minimum threshold of reciprocal overlap ratio between the base and comparison call and it is only applicable on deletions that can be used to evaluate their breakpoint shift; *r* limits the threshold for maximum reference location difference of the compared SVs, which can be used to evaluate the breakpoint shift of insertions. Specifically, in our grid search SV evaluation experiments, we varied *p*, *P*, *O* from 0 to 1 in increments of 0.1, and *r* from 0 to 1 kb in increments of 100 bp.

We first evaluated deletion calls on Hifi_L1. Parameters *p* and *O* had the greatest effects. As depicted in Fig. 3a and b, when *p* and *O* increased, a more stringent correspondence was required between the call and the gold standard to be accepted as a true positive, and consequently, the F1 scores of all SV callers decreased. Moreover, the F1 heatmap and its gradient changes revealed different patterns of performance change in SV callers when more stringent thresholds were applied. The performance of most read alignment-based SV callers, except for pbsv, dropped drastically for values of *p* and *O* greater than 0.7, with smaller than 5% F1 scores when *p* or *O* was set to 1.0. To the contrary, pbsv and the three assembly-based SV callers, Dipcall, SVIM-asm, and PAV, demonstrated stable performance across the grid search, and even maintained a greater than 69% F1 score when an exact match in terms of breakpoints and SV sequence was required (*p* and *O* equal to 1). DeBreak's performance deviated from the rest of methods. Its performance was very sensitive to the changes of *p* but remained robust to changes of *O*. We also investigated the effect of other pairs of parameters on deletion evaluation: *P-r*, *O-r*, *p-P*, *p-r*, and *P-O* (Supplementary Figs. 2–6). Parameters *P* and *r* had little effect on deletion evaluation for all SV callers, unless they were set to 1.0 or 0 bp, respectively. The result indicates that called deletions often have a similar size compared with benchmark deletions, even though their breakpoints may be shifted in most read alignment-based SV callers.

In terms of insertion evaluation on Hifi_L1, *p*, and *r* were chosen as a representative pair to demonstrate the performance change of 12 SV callers. Six read alignment-based SV callers, PBHoney, NanoSV, Smartie-sv_aln, NanoVar, MAMnet, and DeBreak, do not provide alternate allele sequences by default, we thus set *p* to 0 to disable the sequence comparison (Fig. 3e). Compared with deletions, all SV callers exhibited higher sensitivity to a variety of parameters for insertions. As shown in Fig. 3e, f, F1 scores of most tools dropped substantially when *p* was over 0.5, which means the required percentage of allele sequence similarity between the called SV and benchmark was over 0.5. Regarding the parameter *r*, F1 scores decreased as the value of *r* dropped. Once *r* decreased to 200 bp or lower, which means the acceptable reference distance between the SV call and the benchmark was 200 bp or less, F1 scores of all tools decayed markedly. Similarly to what we observed for deletions, assembly-based SV callers and pbsv were more robust to stringent parameters than the rest read alignment-based SV callers. However, even for these robust SV callers, the performance of insertion detection deteriorated to less than 60% F1 score when *r* was set to 0, and less than 40% F1 score when *p* was set

**Table 3 | The user recommendation table**

| DEL and INS ($p = 0$ $P = 0.5$ $r = 500$ $O = 0$) (Pacbio) | size range | SV type | top1 | top2 | top3 | top4 | top5 |
|---|---|---|---|---|---|---|---|
| | 50 bp–1 kb | DEL | cuteSV | Sniffles2 | MAMnet | Sniffles | SVIM |
| | | INS | SKSV | INSnet | MAMnet | PAV | cuteSV |
| | 1 kb–10 kb | DEL | cuteSV | PAV | DeBreak | SKSV | Smartie-sv_asm |
| | | INS | PAV | SKSV | Smartie-sv_asm | SVIM-asm | pbsv |
| | ≥10 kb | DEL | SKSV | cuteSV | SVIM | SVIM-asm | PAV |
| | | INS | Dipcall | PAV | DeBreak | SVIM-asm | Smartie-sv_asm |
| | ≥50 bp | DEL | cuteSV | DeBreak | Sniffles | SKSV | MAMnet |
| | | INS | SKSV | INSnet | PAV | MAMnet | cuteSV |
| | coverage | SV type | top1 | top2 | top3 | top4 | top5 |
| | 5× | DEL | MAMnet | pbsv | PBHoney | Sniffles2 | Smartie-sv_aln |
| | | INS | MAMnet | Sniffles2 | pbsv | Smartie-sv_aln | NanoSV |
| | 10× | DEL | MAMnet | pbsv | PBHoney | Sniffles2 | Smartie-sv_aln |
| | | INS | MAMnet | Sniffles2 | Smartie-sv_aln | Smartie-sv_asm | PAV |
| | 20× | DEL | DeBreak | Sniffles2 | MAMnet | PAV | pbsv |
| | | INS | DeBreak | MAMnet | PAV | Sniffles2 | SVIM-asm |
| | ≥30× | DEL | DeBreak | MAMnet | Sniffles2 | PAV | cuteSV |
| | | INS | DeBreak | MAMnet | Sniffles2 | INSnet | PAV |
| **Complex SV (TRA, INV and DUP) (Pacbio)** | data type | SV type | top1 | top2 | top3 | top4 | top5 |
| | Simulation data (Hifi/CLR) | TRA | NanoSV/cuteSV | cuteSV/pbsv | pbsv/NanoSV | SVIM/SVIM | SKSV/NanoVar |
| | | INV | NanoVar/Sniffles2 | SVIM/cuteSV | cuteSV/SVIM | Sniffles/NanoVar | Sniffles2/Sniffles |
| | | DUP | pbsv/pbsv | DeBreak/NanoVar | NanoVar/Sniffles2 | Sniffles2/DeBreak | SVision/NanoSV |
| | data type | SV type | top1 | top2 | top3 | top4 | top5 |
| | Real cancer data (CLR) | TRA | pbsv | cuteSV | Sniffles | Sniffles2 | SVIM |
| | | INV | pbsv | SVIM | Sniffles2 | SVIM-asm | DeBreak |
| | | DUP | DeBreak | Sniffles2 | Sniffles | cuteSV | pbsv |
| **DEL and INS ($p = 0$ $P = 0.5$ $r = 500$ $O = 0$) (ONT)** | size range | SV type | top1 | top2 | top3 | top4 | top5 |
| | 5 bp–1 kb | DEL | PAV | cuteSV | SVIM | SVIM-asm | SVision |
| | | INS | MAMnet | DeBreak | cuteSV | Sniffles2 | PAV |
| | 1 kb–10 kb | DEL | DeBreak | PAV | Sniffles | Sniffles2 | cuteSV |
| | | INS | DeBreak | cuteSV | PAV | MAMnet | Sniffles2 |
| | ≥10 kb | DEL | SVision | Sniffles | cuteSV | DeBreak | Sniffles2 |
| | | INS | DeBreak | cuteSV | PAV | MAMnet | SVision |
| | ≥50 bp | DEL | PAV | cuteSV | SVIM | SVIM-asm | SVision |
| | | INS | cuteSV | MAMnet | DeBreak | Sniffles2 | PAV |
| | coverage | SV type | top1 | top2 | top3 | top4 | top5 |
| | 5× | DEL | Sniffles2 | NanoVar | PAV | Smartie-sv_aln | DeBreak |
| | | INS | Sniffles2 | PAV | NanoVar | NanoSV | DeBreak |
| | 10× | DEL | Sniffles2 | DeBreak | NanoVar | SVIM-asm | SVision |
| | | INS | Sniffles2 | DeBreak | SVIM-asm | SVision | NanoSV |
| | 20× | DEL | PAV | Sniffles2 | SVision | MAMnet | SVIM-asm |
| | | INS | MAMnet | DeBreak | Sniffles2 | PAV | SVIM-asm |
| | ≥30× | DEL | PAV | SVIM-asm | Sniffles2 | cuteSV | SVIM |
| | | INS | MAMnet | DeBreak | Sniffles2 | PAV | cuteSV |
| **Complex SV (TRA, INV, and DUP) (ONT)** | data type | SV type | top1 | top2 | top3 | top4 | top5 |
| | Simulation data (ONT) | TRA | cuteSV | NanoSV | SVIM | NanoVar | SVIM-asm |
| | | INV | NanoVar | SVIM | cuteSV | Sniffles2 | Sniffles |
| | | DUP | NanoVar | SVision | Sniffles2 | DeBreak | cuteSV |
| | data type | SV type | top1 | top2 | top3 | top4 | top5 |
| | Real cancer data (ONT) | TRA | NanoSV | Sniffles2 | NanoVar | SVIM-asm | cuteSV |
| | | INV | NanoSV | Sniffles2 | Debreak | SVIM-asm | NanoVar |
| | | DUP | Debreak | NanoVar | Sniffles2 | SVision | NanoSV |

**Table 3 (continued) | The user recommendation table**

| Overall performance across datasets for DEL and INS ($p$ = 0 $P$ = 0.5 $r$ = 500 $O$ = 0) | Data type | SV type | top1 | top2 | top3 | top4 | top5 |
|---|---|---|---|---|---|---|---|
| | Hifi | DEL | DeBreak | Sniffles2 | PAV | SVIM-asm | pbsv |
| | | INS | DeBreak | PAV | INSnet | MAMnet | SVIM-asm |
| | CLR | DEL | cuteSV | DeBreak | Sniffles2 | pbsv | Sniffles |
| | | INS | DeBreak | MAMnet | cuteSV | SVIM-asm | INSnet |
| | Nano | DEL | cuteSV | SVIM | Sniffles2 | MAMnet | SVIM-asm |
| | | INS | cuteSV | MAMnet | DeBreak | Sniffles2 | INSnet |
| | **Data type** | **SV type** | **top1** | **top2** | **top3** | **top4** | **top5** |
| | Hifi gt | DEL | PAV | Sniffles2 | SVIM-asm | pbsv | cuteSV |
| | | INS | PAV | SVIM-asm | cuteSV | Sniffles2 | DeBreak |
| | CLR gt | DEL | cuteSV | Sniffles2 | pbsv | DeBreak | SVIM |
| | | INS | cuteSV | DeBreak | SVIM-asm | PAV | Sniffles2 |
| | Nano gt | DEL | cuteSV | Sniffles2 | SVIM | SVIM-asm | DeBreak |
| | | INS | cuteSV | Sniffles2 | DeBreak | SVIM-asm | PAV |

For each evaluation scenario, the table lists several fine-grained conditions and the top 1–5 methods.

to 1.0. We also investigated the effect of other pairs of parameters on insertion evaluation: $p$-$P$ and $P$-$r$ (Supplementary Figs. 7, 8). The parameter $O$ was not used since it is not applicable to insertion evaluation. The F1 heatmap results by grid search experiments show parameter $P$ only had a significant effect on insertion evaluation for all SV callers when it was set to 1.0. This result was consistent with its effect on deletion evaluation.

To reveal why alignment-based SV callers (except pbsv and DeBreak) were more sensitive to changes of parameters compared to assembly-based SV callers, we further analyzed the distribution of SV breakpoint shift and alternate allele sequence similarity for several representative tools (Fig. 3c, d and g, h). The breakpoint shift was calculated from the maximum reference location difference of the two compared SV calls i.e., the maximum start/end location difference between true positive SVs and their corresponding benchmark SVs. The SV sequence similarity was calculated from the edit distance of the two compared SV calls and it was directly extracted from Truvari. The results showed that tools with higher robustness such as PAV and pbsv had a near zero breakpoint shift and near 100% SV sequence similarity with the benchmark callset, compared with tools such as cuteSV and Sniffles, which were very sensitive to the change of parameters and displayed a wide range distribution of breakpoint shift and SV sequence similarity. The result reflects the fact that capturing precise SV breakpoints and alternate allele sequences could establish the tool's robustness under stringent SV evaluation circumstances. DeBreak had a near zero breakpoint shift but a wide range sequence similarity, possibly due to a local assembly strategy employed to better detect breakpoints of SVs combined with a read alignment-based signature to detect the alternative sequence.

We finally performed grid search SV evaluation experiments on Nano_L1. All 9 read alignment-based SV callers, NanoSV, Smartie-sv_aln, Sniffles, SVIM, NanoVar, cuteSV, Sniffles2, MAMnet, and DeBreak showed similar trends with the change of parameters as they showed on Hifi_L1 (Supplementary Fig. 9–17).

In addition to the SV benchmarking tool Truvari, we also used hap-eval[36] to investigate the robustness of all tools across evaluation parameters using grid search. Instead of evaluating each SV independently with Truvari, hap-eval evaluates multiple SVs together (Supplementary Fig. 18). We observed similar patterns and robustness for most of the tools in grid search experiments using hap-eval as in previous grid search experiments using Truvari. Three assembly-based tools (Dipcall, SVIM-asm, and PAV) and one alignment-based tool (pbsv) exhibited significantly higher F1 scores than other SV callers under the most stringent criteria (Supplementary Fig. 19 and Supplementary Fig. 20). In general, insertion SVs were more sensitive to

changes of parameters. More details for these grid search results can be found in supplementary notes section 2.4.

**Orthogonal SV validation with a new complete human genome reference (T2T-CHM13) and trio-based Verkko HG002 assembly**

Although benchmarking against the GIAB SV gold standard is an efficient and precise procedure to evaluate and compare the SV calling performance of different tools, the GIAB gold standard SV callset is not a complete set and could also contain false positives. Relying on the conjecture that SVs supported by more tools are more likely to be true positives than SVs supported by fewer tools, we analyzed overlapping SV calls among 11 read alignment-based or 4 assembly-based SV calling methods, and separated them into three categories by comparing them with the benchmark callset: true positives (TPs), false positives (FPs), and false negatives (FNs). NanoSV is not applicable to this analysis. The detailed method is described in the Methods section. To use benchmark SVs, high-confidence SVs ($N$ = 9397) determined in high-confidence regions by GIAB are often utilized by the community. Without these constraints, we can also consider all SVs from the benchmark ($N$ = 28745). The SV overlapping analysis with constraints on Hifi_L1 showed most of TP SV calls are supported by most of the tools (Fig. 4a, b, top panels). Assembly-based tools are better at detecting TP insertions than read alignment-based tools. More details can be found in supplementary notes section 2.5. Although most of TP SVs were supported by most SV callers, many tools generated a substantial number of exclusive TP SV calls, especially for insertions. An appropriate SV merging and filtering strategy by using SV calls from several robust tools could potentially yield dramatically improved results. More details can be found in supplementary notes section 2.6.

In contrast to true positives, we observed most of FP SV calls were supported by only one or a few tools (Fig. 4c, top panels). We observed that read alignment-based SV callers, PBHoney, Smartie-sv_aln, and NanoVar, generated the most unique deletions and insertions that were not detected by any other tool. Assembly-based SV callers generated much fewer unique SVs, except Smartie-sv_asm (Fig. 4d, top panels). We also performed this SV overlapping analysis without high-confidence constraints (Fig. 4a–f, bottom panels), and found overlapping percentage results for all tools were similar to those with constraints, except higher absolute numbers were shown in both TPs and FPs when all benchmark SVs were used. However, for FN SV calls (Fig. 4e–f, bottom panels), the pattern was different with constraints versus without constraints. Without constraints, most FN calls could not be detected by most tools. This result reveals the reason why GIAB does not consider most of them as high-confidence SVs, as it is difficult to determine whether they are real true positives or not.

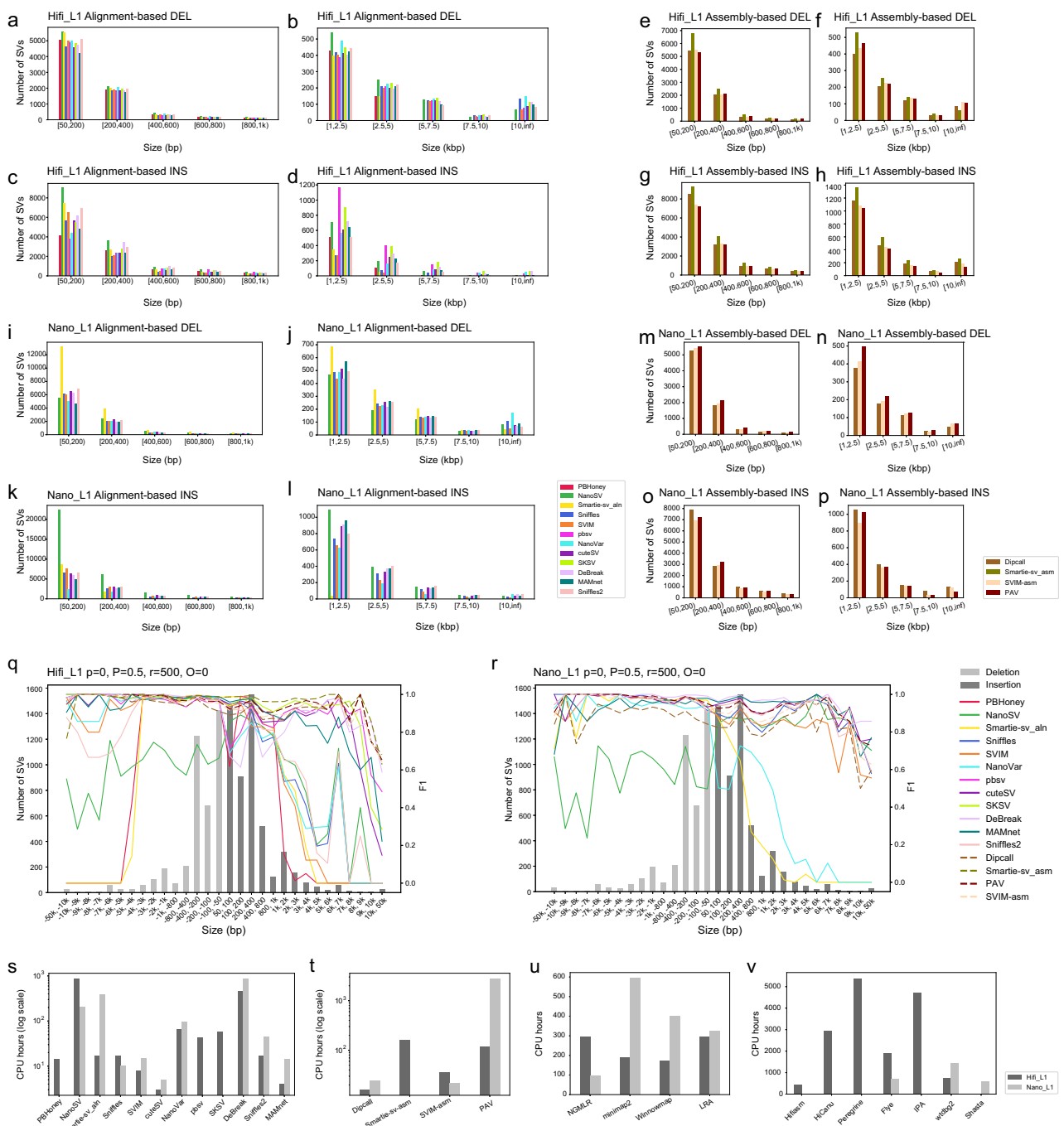

**Fig. 2 | Size distribution, accuracy, and CPU time consumption for SV discovery in Hifi_L1 and Nano_L1. a–d** Deletion and insertion SV size distribution (50 bp–1 kb and ≥1kb) discovered by read alignment-based tools on Hifi_L1. **e–h** Deletion and insertion SV size distribution (50 bp–1 kb and ≥1kb) discovered by assembly-based tools on Hifi_L1. **i–l** Deletion and insertion SV size distribution (50 bp–1 kb and ≥ 1 kb) discovered by read alignment-based tools on Nano_L1. **m–p** Deletion and insertion SV size distribution (50 bp–1 kb and ≥1 kb) discovered by assembly-based tools on Nano_L1. **q, r** F1 accuracy of SV detection at different size ranges by tuning four different combinations of evaluation parameters. Negative size range

represents deletion SVs, and positive size range represents insertion SVs. Bar plot shows benchmark SV distribution at different size ranges. The line plot shows the F1 score of 16 different SV calling methods. *O* is the minimum reciprocal overlap between SV call and gold standard SV. *p* is the minimum percentage of allele sequence similarity. *r* is the maximum reference location distance between SV call and gold standard SV. *O* and *p* vary from 0–1 with a 0.1 interval, while *r* varies from 0–1000 bp with a 100 bp interval. **s–v** CPU time consumption for read alignment-based and assembly-based SV callers, different aligners, and assemblers. Source data are provided as a Source Data file.

We, therefore, wished to apply another orthogonal SV validation approach to evaluate those FP and FN SV calls (without constraints). We used the new complete sequence of human genome T2T-CHM13[37] and the trio-based Verkko HG002 assembly to investigate whether those calls can be supported or not. Verkko uses the full-coverage HG002 dataset (105 × HiFi and 85 × ONT UL) to produce the most

continuous assembly of this genome to date[38]. The detailed method is described in the Methods section. We ordered tools by the number of FP calls and observed that all 15 methods had FPs supported by T2T-CHM13 or Verkko assembly and these calls could be true positives but missed by GIAB. The four assembly-based methods and two alignment-based methods (NanoVar and Smartie-sv_aln) achieved the best results

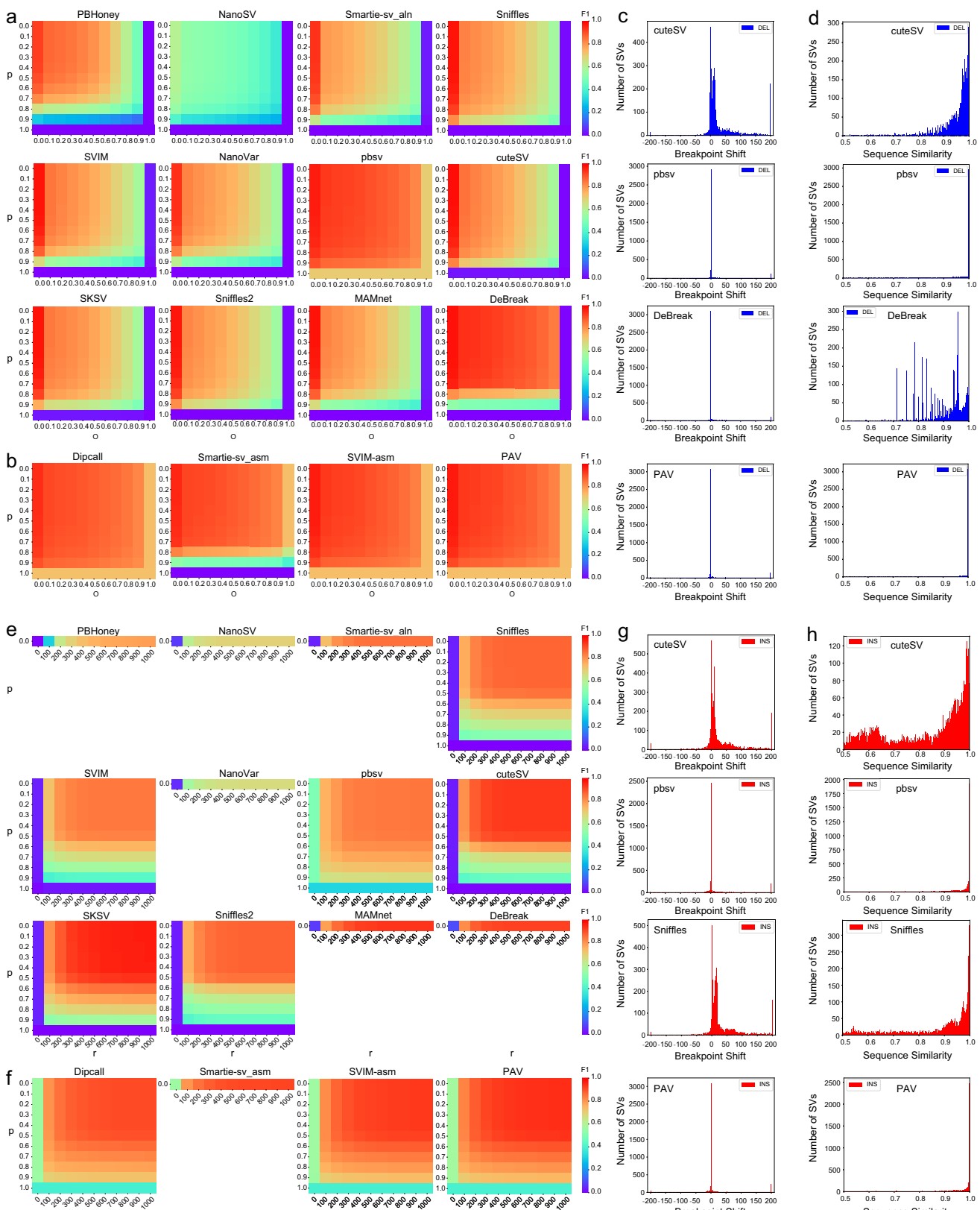

**Fig. 3 | F1 accuracy by tuning different evaluation parameters and distribution of breakpoint shift and alternate allele sequence similarity for SVs on Hifi_L1.** **a** Grid search heatmap of F1 values for deletion SVs by different read alignment-based tools. *O* is the minimum reciprocal overlap between SV call and gold standard SV. *p* is the minimum percentage of allele sequence similarity between SV call and gold standard SV. *O* and *p* vary from 0–1 with a 0.1 interval. **b** Grid search F1 heatmap for deletion SVs by different assembly-based tools. **c** Distribution of breakpoint shift for deletion SVs. **d** Distribution of alternate sequence similarity for deletion SVs. **e** Grid search F1 heatmap for insertion SVs by different read alignment-based tools. *r* is the maximum reference location distance between SV call and gold standard SV. *p* is the minimum percentage of allele sequence similarity between SV call and gold standard SV. *p* vary from 0–1 with a 0.1 interval. *r* varies from 0–1000 bp with a 100 bp interval. **f** Grid search F1 heatmap for insertion SVs by different assembly-based tools. **g** Distribution of breakpoint shift for insertion SVs. **h** Distribution of alternate sequence similarity for insertion SVs. Source data are provided as a Source Data file.

when considering both the absolute number of supported FP deletions and the percentage of supported FP deletions (Fig. 4g). With respect to FP insertions, three assembly-based tools (SVIM-asm, PAV, and Dipcall) achieved twice as many supported FP insertions compared to alignment-based tools, and they also had the largest percentage of supported insertions among their FPs (Fig. 4g). For FPs supported by Verkko assembly, we observed highly concordant patterns for all methods as T2T (Fig. 4i), except that Verkko assembly supported fewer FP deletions than T2T. We also found that 93.7−99.3% FPs of each tool supported by Verkko assembly were also supported by T2T. We highlighted two potential explanations for this observation: (1) T2T-CHM13 provides a more comprehensive assembly compared to Verkko assembly (N50 is 151Mb in T2T-CHM13 compared to 136Mb in the Verkko assembly); (2) The CHM13 genome may be more similar to HG002 in certain regions or contain fewer unique variations, making it easier to detect true positive SVs in HG002 using CHM13-T2T. We also analyzed the SV size distribution of all FPs and FPs supported by T2T-CHM13, and did not observe any significant difference (Fig. 4h). We did observe a significant number of FN SVs supported by T2T-CHM13 or Verkko assembly, even though most of them were non-high-confidence SVs and not supported by most tools, especially for insertions (Supplementary Fig. 21a−d). These non-high-confidence SVs could really be true positives by this orthogonal validation. We demonstrated one 5.4kb FP deletion and one 780 bp FP insertion supported by T2T-CHM13 in IGV (Supplementary Fig. 21e). We next analyzed overlapping SV calls on Nano_L1 among 8 read alignment-based and 3 assembly-based SV callers and observed a similar pattern for overlapping calls among alignment- and assembly-based tools, respectively (Supplementary Fig. 22).

## Subsampling effects on SV calling

In order to assess SV callers' robustness to different sequencing coverages, we subsampled the raw reads of Hifi_L1 and Nano_L1 to lower coverages such as 50×, 40×, 30×, 20×, 10×, and 5× fold using rasusa[39], and evaluated the performance of the tools after performing SV calling on these subsampled datasets (Fig. 5, Supplementary Fig. 23, and Supplementary Table 8-10). To perform this analysis, the same set of fixed and moderate-tolerance parameters was used ($p = 0$, $P = 0.5$, $r = 500$, $O = 0$).

We first analyzed the subsampling effects of 12 read alignment-based SV calling methods on deletions in Hifi_L1 to identify tools with most sustained performance (Fig. 5a−c). As the sequencing coverage decreased from 50× to 20×, the deletion recall of tools that rely on a similar "signature detection-clustering-genotyping" pipeline, including Sniffles, SVIM, cuteSV, and SKSV, dropped drastically (Fig. 5a−b and Supplementary Table 8, the solid line segments with star markers denote 20×). The deletion recall of these four tools further dropped to approximately 1% at 5× fold coverage, while still maintaining high precision (the solid line segments with circle markers denote 5x). The performance of NanoSV was already low even at 50×. On the other hand, PBHoney, NanoVar, Smartie-sv_aln, pbsv, Sniffle2, MAMnet, and DeBreak demonstrated relatively high and stable performance across different sequencing coverages (Fig. 5a−c). Among these robust tools, pbsv and MAMnet achieved the highest F1 (93.4% and 94.0%), and DeBreak had the lowest F1 (60.7%) at 5x. With respect to the effects of low sequencing coverage on insertions (Fig. 5d−f and Supplementary Table 8), we observed similar subsampling effects, except that recall and precision scores were generally lower in insertions than in deletions.

We also investigated the subsampling effects on SV calling on Nano_L1 (Supplementary Fig. 23a−d and Supplementary Table 9). Similar to what was observed in Hifi_L1, the deletion and insertion recall of Sniffles, SVIM, and cuteSV dropped substantially when sequencing coverage decreased to 10−20×, but lower sequencing coverage had less impact on precision. Among alignment-based methods, the performance of Sniffles2 and DeBreak was relatively

high and robust to the changes of sequencing coverage in both deletion and insertion calling. MAMnet also achieved high and robust performance at 20−40×, but could not output results on low coverage (5−10×) data. The performance of NanoSV, NanoVar, and Smartie-sv_aln was relatively low, even at 40×, especially for insertions. However, their performance was relatively robust and maintained better F1 values at low (5−10×) coverage than most of the tools.

We further evaluated the effects of subsampling on SV calling, using assembly-based SV callers with subsampled Hifi_L1 data (Fig. 5g−h and Supplementary Table 10). In this analysis, all four assembly-based SV callers accepted the assembly result of Hifiasm as input. For both deletions and insertions, unlike the monotonic decrease of recall as a function of coverage in most read alignment-based SV callers, there were surprisingly few changes in the performance of assembly-based SV callers (with the exception of Dipcall; the solid line segments with triangle markers denote 10×), until the sequencing coverage dropped below 10 fold. We also observed similar high and robust performance for the three assembly-based tools on subsampled Nano_L1 (Supplementary Fig. 23c−d and Supplementary Table 11).

Lastly, we analyzed the subsampling effects on genotyping (gt) accuracy for all robust tools, except PBHoney and Smartie-sv_aln, which do not output genotyping results (Fig. 5a−h and Supplementary Fig. 23a−d, dashed vs. solid lines, and Supplementary Tables 8-11). In general, the genotyping recall, precision, and F1 scores showed similar trends with overall accuracy. However, the genotyping performance of the three robust assembly-based tools (Smartie-sv_asm, svim-asm, and PAV), plus DeBreak and MAMnet declined considerably when the sequencing coverage decreased to 5−10×. In contrast, the robust alignment-based tools, pbsv, and Sniffles2 exhibited much better genotyping performance at low coverage. Low assembly continuity and high assembly break at low sequencing coverage (5−10×) could be the reason the genotyping accuracy of assembly-based tools decreased significantly.

## SV calling performance across 11 PacBio and ONT datasets

To further evaluate the robustness of each tool across different libraries, we performed analysis across 11 long-read datasets. The same set of fixed and moderate-tolerance parameters ($p = 0$, $P = 0.5$, $r = 500$, $O = 0$ for Truvari) was selected to evaluate the performance of all tools. The parameter $p$ was set to zero to disable the SV sequence comparison since five tools do not provide alternate allele sequences for insertions, as mentioned before. The parameter $O$ was also set to zero to allow breakpoint shifts for deletions for most alignment-based tools. Across five Hifi long reads datasets (Fig. 5i), all four assembly-based tools achieved consistently high F1 scores for both deletion and insertion SVs regardless of different coverage (28−56.3×) or insert sizes (10−20 kb) in different libraries (Table 1). Four alignment-based tools (cuteSV, Sniffles2, MAMnet, and DeBreak) also achieved consistently high performance across five Hifi datasets for both deletion and insertion SVs. PAV maintained the highest genotyping accuracy. The genotyping accuracy of insertions for all robust alignment-based tools decreased markedly compared to their overall accuracy, particularly in MAMnet. For the rest of the alignment-based tools, overall accuracy and genotyping accuracy of insertion SVs varied considerably across different datasets. For the three CLR and three ONT datasets, all four alignment-based tools (cuteSV, Sniffles2, MAMnet, and DeBreak) maintained high and consistent F1 scores. The genotyping accuracy of MAMnet was much lower than its overall accuracy. Across all CLR and ONT datasets, only one of the assembly-based tools, SVIM-asm, achieved performance at par with the robust alignment-based tools (cuteSV, Sniffles2, and DeBreak).

## Effects of different aligners and assemblers

Considering that different aligners could affect the performance of read alignment-based SV callers, we selected three read alignment-

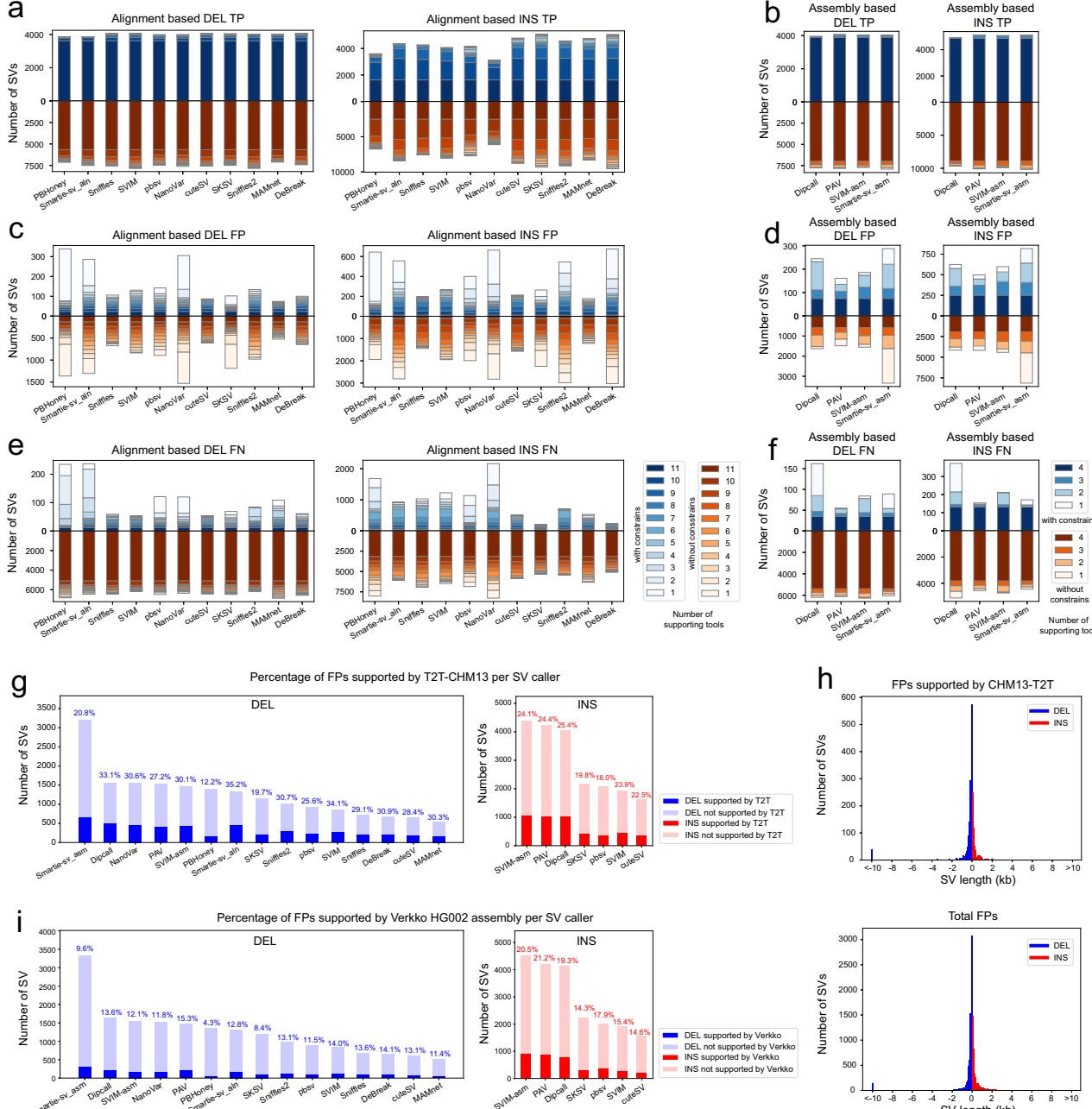

**Fig. 4 | Orthogonal SV validation with overlapping calls among different tools and with the new complete human genome reference T2T-CHM13.**
**a**, **b** Overlapping TP deletion and insertion SVs supported by read alignment-based and assembly-based tools. **c**, **d** Overlapping FP deletion and insertion SVs supported by read alignment-based and assembly-based tools. **e**, **f** Overlapping FN deletion and insertion SVs supported by read alignment-based and assembly-based tools. The top panels show results by using high-confidence benchmark SVs from GIAB (with constraints), and the bottom panels show results by using all benchmark SVs (without constraints). The height of each bar represents the total SVs discovered by a tool. Color gradient in the bar specifies a range of supporting tools for SVs. Dark colors represent a high number of supporting tools, while light colors represent a low number of supporting tools. **g** Percentage of FP deletions and insertions supported by T2T-CHM13 for each SV caller. **h** Size distribution of total FPs (without constraints) and FPs supported by T2T-CHM13. **i** Percentage of FP deletions and insertions supported by Verkko assembly for each SV caller. Source data are provided as a Source Data file.

based tools, Sniffles, SVIM, and cuteSV, which had relatively higher performance on both Hifi_L1 and Nano_L1, and evaluated the effect of four different aligners (NGMLR[20], minimap2[35], Winnowmap[40], and LRA[41]) on their SV calling performance. We used radar plots to demonstrate the SV calling performance of three tools across four metrics (Fig. 6a).

In general, although the effect may be mitigated by optimizing SV calling algorithms, using different aligners did affect SV calling performance, especially for insertions (Fig. 6). Minimap2, Winnowmap,

and LRA generally performed best for insertion recall. Performance on deletions was relatively unaffected by different aligners. CuteSV was least affected by different aligners. More details are provided in supplementary notes section 2.7.

We similarly evaluated the effects of six different assemblers on Hifi_L1 and three different assemblers on Nano_L1, using three assembly-based SV callers (Fig. 6b). In general, for Hifi data, assembly results from Hifiasm achieved the best SV calling, followed by Peregrine + HapDup and wtdbg2 + HapDup. Deletion and insertion

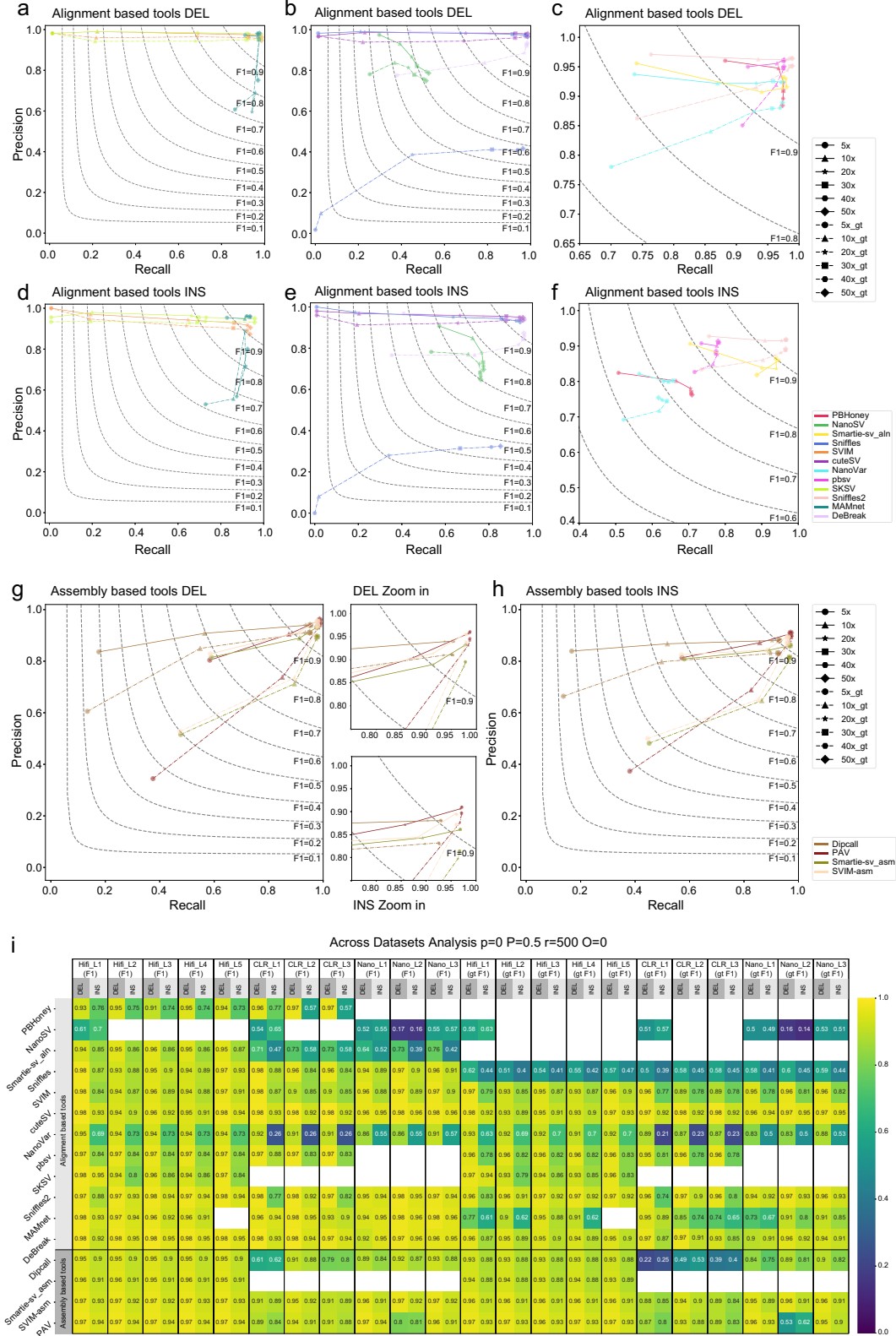

**Fig. 5 | Subsampling effect and across-datasets analysis of different SV callers.**
**a–f** Recall-precision-F1 curves show the subsampling effect on deletion and insertion SVs by read alignment-based tools on Hifi_L1. **g–h** Recall-precision-F1 curves show the subsampling effect on deletion and insertion SVs by assembly-based tools on Hifi_L1. The coverage depth varies from 5x, 10×, 20×, 30×, 40× to 50×. Solid lines with markers are for different coverage depths, and corresponding dashed lines are for genotyping (gt) accuracy. For deletion SVs, we zoom in on the top right part of the plot to demonstrate the curves more clearly. **i** Heatmap shows overall and genotyping (gt) F1 scores on 11 long reads datasets for 16 SV calling methods. Empty cells represent analysis that could not be performed (or finished within 14 days of runtime) for the tool in the corresponding row. Source data are provided as a Source Data file.

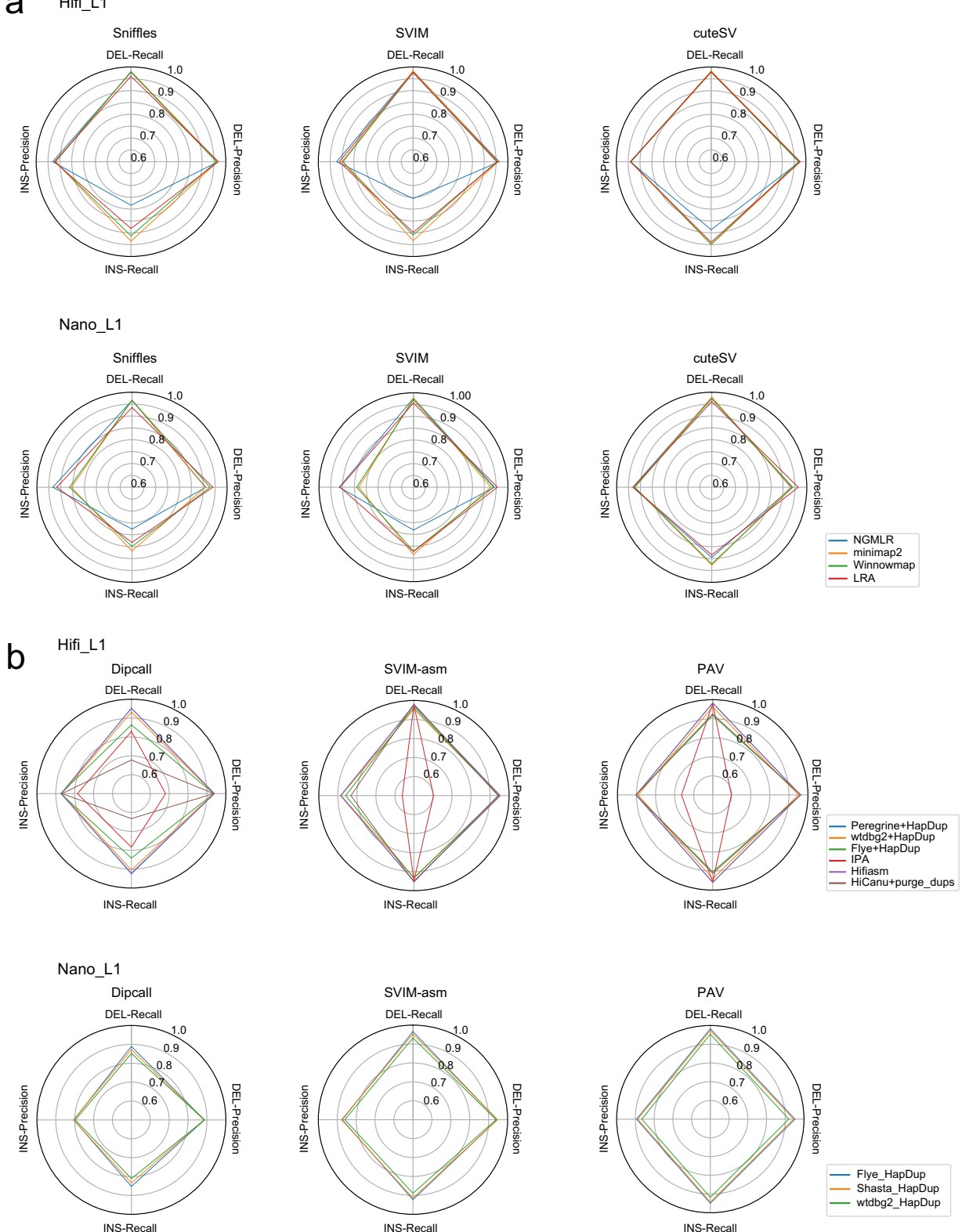

**Fig. 6 | The effects of different aligners and assemblers on SV calling of read alignment-based and assembly-based tools, respectively. a** The effect of different aligners on SV calling of read alignment-based tools for Hifi_L1 and Nano_L1. **b** The effect of different assemblers on SV calling of assembly-based tools for Hifi_L1 and Nano_L1. Source data are provided as a Source Data file.

precision were poor when using contigs from the IPA assembler. Dipcall was most affected by the choice of assembler. The choice of assembler was more subtle on ONT than Hifi data. More details are provided in supplementary notes section 2.8.

## Evaluation of complex SVs with simulated datasets

Large deletions and insertions account for most SVs, but other SVs, such as translocations, inversions, and duplications, also describe different combinations of DNA rearrangements. The lack of

benchmarking data for such complex SVs makes it difficult to evaluate tools. We thus simulated 9 long reads datasets to benchmark translocations, inversions, and duplications. Simulation and evaluation details are outlined in the Methods section. In this analysis, we only included ten SV calling methods that can detect complex SVs (Fig. 7a).

Alignment-based tools performed well with these types of complex SVs. For translocations, cuteSV achieved the best overall performance for PacBio Hifi (F1 = 96%), CLR (F1 = 96%), and ONT (F1 = 97%) datasets, followed by NanoSV and SVIM. pbsv performed equally well with cuteSV on the PacBio Hifi and CLR datasets, but was not designed to detect translocations on the ONT dataset. SKSV achieved a 94% F1 score on the Hifi dataset, but was not designed to detect translocations on the CLR and ONT datasets. Performance across all tools was fairly poor for inversions: NanoVar achieved the best overall performance for Hifi (F1 = 75%) and ONT (F1 = 67%), followed by SVIM. Sniffles2 achieved the best performance for CLR (F1 = 67%), followed by SVIM. Finally, for duplications, pbsv reached a 92% F1 score for Hifi. However, all tools performed relatively poorly on CLR and ONT datasets. pbsv still achieved the best F1 (66%) for the CLR dataset; it was not designed to detect duplications in the ONT dataset. NanoVar achieved the best F1 (62%) for ONT. In summary, inversion or duplication detection was not as good as other SV types (translocations, deletions, and insertions). The six best tools, cuteSV, NanoSV, SVIM, pbsv, NanoVar, and Sniffles2 had their own strengths and weaknesses in complex SV calling across different long reads datasets.

The genotyping accuracy of these six high-performing tools in inversions and duplicate calling decreased in the range of 0–46.3%. SVIM had the minimum genotyping accuracy drop in the range of 0–1.9%. The maximum genotyping F1 drop reached 46.3% in cuteSV, 39.3% in Sniffle2, 31.1% in NanoSV, 20.8% in pbsv, and 13.6% in NanoVar. Notably, Sniffles had extremely low genotyping accuracy in all Hifi, CLR, and ONT datasets since it assigned most of its complex SVs with the wrong genotyping and had a bias to call all homozygous SVs to be heterozygous ones. Its low genotyping accuracy was also reflected in both deletion and insertion SVs.

### SV calling performance on cancer datasets

To further evaluate the performance of the tools in detecting complex SVs in real data, we performed additional analysis on two publicly available sets of tumor-normal paired libraries (Pacbio CLR and ONT) and investigated five classes of somatic SVs. Talsania et al.'s work[42] provided a high-confidence HCC1395 somatic SV callset that we used for benchmarking. Details for somatic SV calling and evaluation are described in the Methods section. The high-confidence HCC1395 somatic SV callset has a total of 1777 SVs, including 551 insertions, 717 deletions, 146 translocations, 133 inversions, and 230 duplications. Since this high-confidence callset is incomplete, we plotted recall and precision separately. We expected the precision scores of most of the tools to be low and thus only focused on recall to compare the somatic calling results (Fig. 7b).

Performance in this real dataset was generally consistent with the simulation results, for most tools. For somatic translocations, pbsv and NanoSV achieved the best performance in the CLR or ONT datasets, respectively (recall = 55% for CLR in pbsv; recall = 53% for ONT in NanoSV), having ranked second-best in the simulation experiments. For somatic inversions, pbsv achieved the best recall (48%) for CLR, followed by SVIM and Sniffles2, which were the two best tools to call inversions in simulated CLR data. NanoSV achieved the best recall (41%) for ONT. However, NanoVar did not perform as good an inversion calling as it did in the simulation experiments. For somatic duplication evaluations, DeBreak achieved the best recall: 75% for CLR and 54% for ONT, respectively. Tools such as NanoVar that performed well in duplication calling for simulation experiments also performed relatively well in real datasets (43% for CLR and 53% for ONT).

### Benchmarking most recent deep learning-based SV calling framework

Recently, more deep learning-based SV calling methods for long reads have been introduced, including SVision and INSnet. The use of advanced deep-learning methods promises superior SV detection compared to traditional alignment-based methods. These tools appeared in the course of developing this article, and we were able to benchmark them rapidly, using our established evaluation framework and to compare them with previously evaluated tools (Supplementary Figs. 24, 25, and Supplementary Tables 12, 13). Overall, SVision and INSnet achieved fairly reasonable and robust performance on deletion and insertion SV calling, however, neither of them achieved superior performance for most of the scenarios or conditions we benchmarked when comparing with previously evaluated tools. More details for these benchmarking results can be found in the supplementary notes section 2.9. For some conditions, SVision and INSnet could be optimal tools. They were also included in our discussion and user recommendations. With our established benchmarking framework, future tools can be easily added for a comprehensive comparison.

## Discussion

In this study, to comprehensively compare alignment-based (including the hybrid and deep learning-based) and assembly-based SV callers, we first analyzed SV calling performance under a set of moderate-tolerance parameters in the HG002 sample, relative to the GIAB SV gold standard callset. Our main results were as follows: a) Assembly-based tools detected more large insertions than most alignment-based tools, especially those greater than 1 kb. b) Detection accuracy of assembly-based tools was more robust to SV size changes than most alignment-based tools. Among alignment-based tools, SKSV, cuteSV, MAMnet were more robust to changes in SV size. c) Including the assembly procedure, assembly-based SV calling pipelines took much longer CPU time to finish compared to alignment-based SV calling pipelines, which involved a time-efficient alignment process. Based on these results we call attention to the trade-off between computing speed and performance for alignment- and assembly-based tools. Depending on user needs, we provide guidance for performance in Table 3 and Supplementary Table 1 across a series of criteria.

### Performance across parameters and datasets

Due to the complexity of SVs, a set of fixed or moderate-tolerance parameters does not capture the whole characteristics of different SV callers and could cause biases when benchmarking them. We thus designed a set of grid search SV evaluation experiments by tuning parameters affecting SV breakpoint shift and alternate sequence similarity between the called SVs and the benchmark SV set, to assess the performance and robustness of different SV callers. These results indicated: a) Read alignment-based SV callers, with the exception of pbsv and DeBreak, were more sensitive to parameter changes. DeBreak showed high accuracy in terms of breakpoint shift, though lower accuracy in terms of sequence similarity. b) Three assembly-based tools (Dipcall, SVIM-asm, and PAV) and one alignment-based tool (pbsv) were robust to parameter changes as they achieved a near zero breakpoint shift and near 100% SV sequence similarity with the benchmark callset.

Although the GIAB SV gold standard callset provides us with an efficient and appropriate way to evaluate SV calls among different tools, this callset is not complete and is likely to contain false positives. Analyzing the overlapping calls among different tools provides an alternative perspective for evaluation. Furthermore, we employed a new complete sequence of human genome T2T-CHM13 and trio-based Verkko HG002 assembly to investigate whether those likely false positive calls of each tool or low-confidence calls from GIAB were supported or not. We observed the following: a) Most TPs in high-confidence regions were supported by most of the tools. b) More than

a

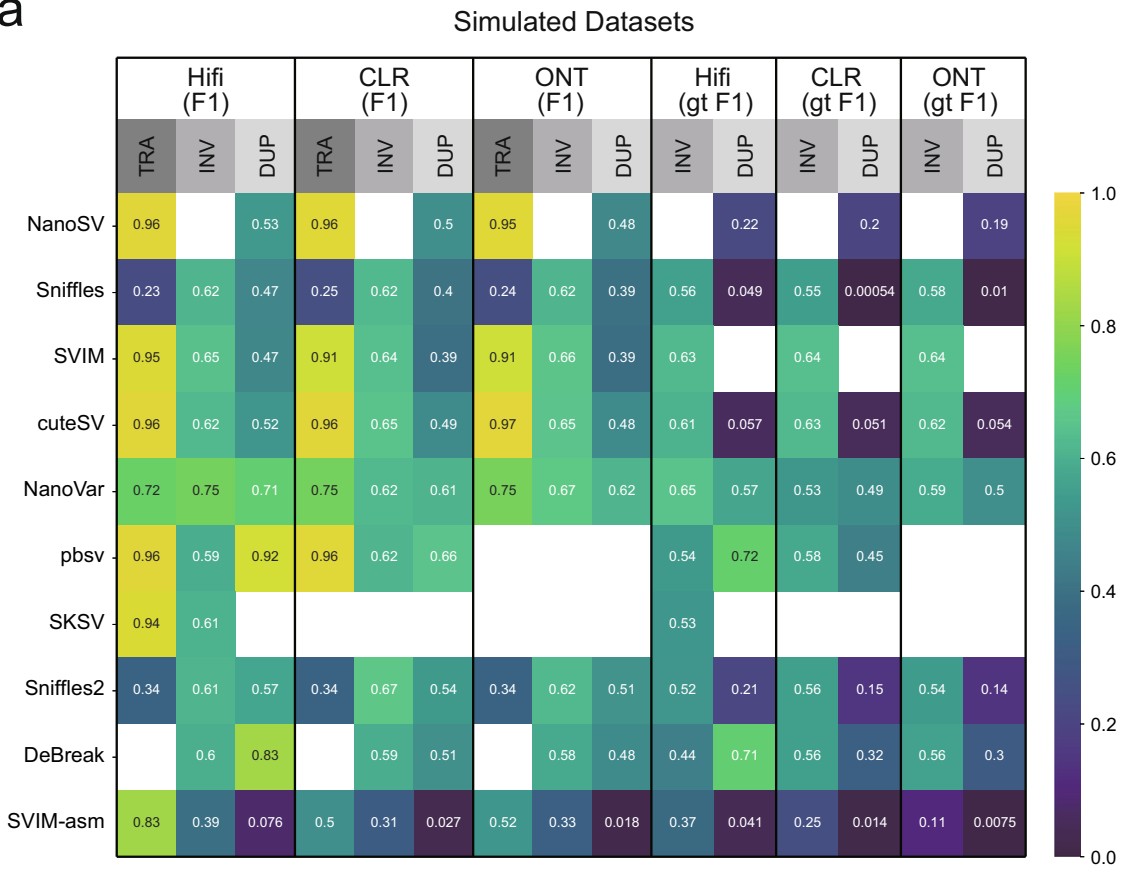

b

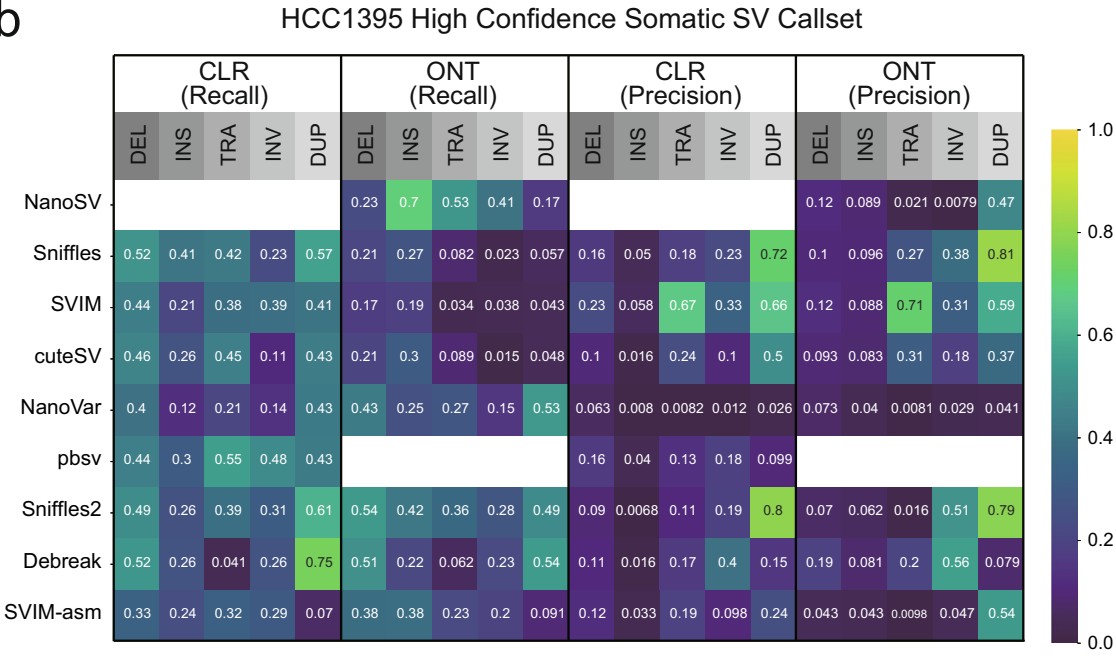

**Fig. 7 | Complex SV detection in simulated and real cancer datasets. a** Heatmap shows overall and genotyping (gt) F1 scores of translocation (TRA), inversion (INV), and duplication (DUP) detection for 10 SV calling methods on 9 simulated PacBio Hifi, CLR, and ONT datasets. **b** Heatmap shows recall and precision scores of somatic deletion (DEL), insertion (INS), translocation (TRA), inversion (INV), and duplication (DUP) detection for 9 SV calling methods on two publicly available sets of Tumor-Normal paired Pacbio CLR and ONT libraries. Empty cells represent analysis that could not be performed (or finished within 14 days of runtime) for the tool in the corresponding row. Source data are provided as a Source Data file.

20% of FP insertions identified by assembly-based tools (Dipcall, SVIM-asm, and PAV) were supported by both T2T-CHM13 and Verkko assembly, indicating that they may be true positives missed by GIAB. c) FNs that appeared in regions that GIAB characterized as low-confidence (without constraints) were rarely supported by the tools we tested, however, some were supported by T2T-CHM13 and Verkko assembly.

We further analyzed the subsampling effects on SV calling. Our conclusions were as follows: (a) Alignment-based tools that integrated signature detection pipelines (Sniffles, SVIM, cuteSV, and SKSV) were very sensitive to sequencing coverage changes. Alignment-based tools (PBHoney, NanoVar, Smartie-sv_aln, pbsv, Sniffle2, MAMnet, and DeBreak) were more robust to coverage changes while maintaining a good performance. Subsampling with lower coverage had a significant effect on recall and F1, but a relatively smaller impact on precision. (b) For assembly-based tools (except Dipcall), decreasing of sequencing coverage had little impact on performance, until it dropped to less than 10×. (c) At low sequencing coverage (5−10×), the genotyping accuracy of all three robust assembly-based tools decreased markedly, but two robust alignment-based tools (pbsv and Sniffles2) still maintained high genotyping accuracy.

In addition to subsampled datasets, we also analyzed SV calling performance across 11 different PacBio and ONT datasets. We found a) All four assembly-based tools (PAV, Dipcall, SVIM-asm, Smartie-SV_asm) and six alignment-based tools (cuteSV, Sniffles2, MAMnet, DeBreak, and INSnet) demonstrated stable and high performance across five Hifi datasets regardless of coverage (28×−56.3×) or insert sizes (10-20 kb). For all three CLR datasets, these alignment-based tools still maintained high and stable F1 scores, except Sniffles2, and INSnet. For all three ONT datasets, these alignment-based tools still maintained high and stable F1 scores. However, only one of the assembly-based tools, SVIM-asm, achieved performance at par with the alignment-based tools on CLR and ONT datasets. b) The genotyping accuracy of insertions for all robust alignment-based tools decreased markedly compared to their overall accuracy, particularly in MAMnet.

Since different aligners and assemblers could influence read alignment-based tools and assembly-based tools, respectively, we analyzed the effects of four aligners and six assemblers. a) Different aligners had a noticeable effect on the insertion recall of alignment-based tools. In general, minimap2, Winnowmap, and LRA were better than NGMLR for most alignment-based tools with respect to insertion recall. b) Insertion precision, deletion recall, and precision were unaffected, or only subtly affected by different aligners. c) Compared to Sniffles and SVIM, cuteSV was influenced the least by different aligners. d) Hifiasm had the best performance on Hifi data, and SV callers showed the worst precision performance on contigs assembled by IPA. e) The performance of Dipcall was greatly influenced by different assemblers. f) The effect of different assemblers was more subtle on ONT data than Hifi data.

We finally investigated complex SVs including translocations, inversions, and duplications in both simulated and paired tumor-normal datasets. In simulation experiments, we found the following: a) For translocations, on simulated data, the overall best tools were cuteSV and pbsv (F1 > 0.95 on all suitable datasets), followed by NanoSV and SVIM. b) For inversions, no tool achieved high performance. The best F1 scores were achieved by NanoVar, Sniffles2, and NanoVar on Hifi, CLR, and ONT, respectively. c) For duplications, pbsv was the best tool for Hifi and CLR data followed by DeBreak, while NanoVar was the best for ONT data. In real datasets, we reached the following conclusions: a) For somatic translocations, pbsv had the best recall on CLR data, and NanoSV on ONT data. This performance was consistent with that in simulated datasets. b) For somatic inversions, pbsv had the highest recall on CLR data, while NanoSV had the highest recall on ONT data. c) For somatic

duplications, DeBreak had the overall best performance on CLR and ONT data.

## Recommendations

Our study provided systematic performance comparisons for read alignment-based (including the hybrid and deep learning-based) and assembly-based SV calling tools. Along with the conclusions we drew from each designed experiment, we also designed a user recommendation table to highlight optimal tools based on 31 fine-grained conditions for both PacBio and ONT data. Top tools with high ranking or robust performance in each condition are summarized in Table 3 and Supplementary Table 1. No tool achieved superior performance across all conditions. Users can select the tools best suited for their needs based on different scenarios. Alignment-based approaches are almost universally used at present because they are less computationally demanding, and more new tools are developed every year. As we see from the table, alignment-based tools are ranked high when a set of moderate-tolerance evaluation parameters are used ($p = 0$ and $O = 0$ are favorable for alignment-based tools). However, such approaches have limitations in the accurate representation of a complete genome, an SV's initial and ending position in the genome (its "breakpoints"), and in identifying the full SV sequence. If users need to apply more rigorous evaluation thresholds, assembly-based tools like PAV could be the best in most conditions, and the ranking of other good alignment-based tools will be reordered depending on their robustness. More deep learning-based approaches (built on read alignment) continue to emerge and generate adequate and robust performance. MAMnet is representative of this trend, and it is ranked high in most conditions, although its genotyping accuracy is often much lower than the overall accuracy. However, deep learning-based methods still suffer from the same problem as traditional alignment-based methods. When users choose the optimal tools based on their needs, they, therefore, need to be aware of the SV evaluation thresholds. Future algorithms need to resolve this problem, and one of the promising approaches may be integrating local assembly strategy through a hybrid method. DeBreak is an example and it achieves good performance in most conditions. Much fewer assembly-based tools have been developed, but assembly-based tools are more efficient and robust in detecting precise SVs, compared to alignment-based methods. Assembly-based tools, on the other hand, are more likely to have low genotyping accuracy at low (5−10×) sequencing coverage due to assembly breaks, and most of the assembly-based tools were not designed to detect complex SVs. Future studies should take into account the importance of diploid genome assembly and assembly-based tools, even though assembling the whole genome of an individual is much more demanding in computational resources. More large-scale and efficient genome assembly algorithms and assembly-based tools are likely to further improve precise SV detection.

## Methods
### Overview of existing SV callers
Existing long-read-based SV callers can be divided into read alignment-based (including deep learning-based and hybrid methods) and assembly-based methods as mentioned above. Within each category, SV callers also vary in their exact SV detection strategies.

In general, read alignment-based methods detect SVs by analyzing discordance between the sample genome's reads and the reference genome based on read alignments. PBHoney, proposed in 2014, parses inter-alignment signals with soft-clipped tails and intra-alignment discordance signals independently. A best re-alignment of each soft-clipped tail along with the initial read alignment composes a piece-alignment, which is used to identify SVs. Regarding intra-alignment discordance, PBHoney first counts three types of errors, mismatch, deletion, and insertion, to produce three corresponding channels. It then identifies possible SV regions that contain increases in

discordance followed by decreases in discordance, corresponding to the starts and ends of genomic variants, respectively.

NanoSV was introduced in 2017. It first processes each read into a sequence of aligned segments and alignment gaps, where each aligned segment is represented by a tuple containing the segment's chromosome, genome start and end coordinates, and the mapping orientation. It then defines each candidate breakpoint junction by any two consecutive aligned segments on a read together with the gap in between. Finally, NanoSV detects SVs by clustering breakpoint junctions.

Sniffles, SVIM, cuteSV, SKSV, and pbsv, which were introduced in 2018, 2019, 2020, 2021, and 2021, respectively, leveraged similar "signature detection-clustering-genotyping" pipelines, and we will therefore refer to them as signature-based methods. For these methods, signatures are sets of information extracted from intra-alignment discordance or inter-alignment discordance of a read, and a signature usually contains the read name, start and end coordinates, the chromosome name, the strand orientation, SV type, and other tailored information specific for each SV type. Although signature-based tools use similar strategies to identify signatures, the clustering strategy varies from tool to tool. Sniffles stores and merges SV signatures using a self-balancing binary tree, where each node collects multiple related signatures to form an SV. Sniffles2, introduced recently, is a redesign of the Sniffles pipeline for higher accuracy and time efficiency, and it also extends to population-scale SV calling. In the clustering step, Sniffles2 implements a three-phase clustering. It first clusters the raw SV signals by their SV type and genome start location, then corrects alignment errors in highly repetitive regions, and finally splits the preliminary clusters to represent different supported SV lengths. SVIM uses a graph-based method to identify signature clusters to form final SVs, where each node corresponds to an SV signature. cuteSV clusters the signatures in two steps, first by their types and genome locations, and then by signature length to identify sub-clusters for different alleles of complex heterozygous SVs. SKSV, introduced by the same group of investigators as cuteSV, directly uses cuteSV to detect SVs from the signatures. However, instead of collecting signatures from the read alignments provided by existing aligners, SKSV detects potential SV signatures by identifying non-co-linear alignment segments using a greedy extension strategy. pbsv clusters signatures according to their length difference and overlapping on the reference genome. Additionally, pbsv generates a consensus flanking region for each cluster, and aligns the cluster together with the consensus flanks back to the reference genome. With regard to the final genotyping step, signature-based tools typically leverage information from supporting reads of each SV.

Diverging from signature-based tools, NanoVar, a deep learning-based approach using read alignment information, proposed in 2020, introduces novel adjacency from split-read or hard-clipped alignments. A novel adjacency is defined as two adjacent genomic coordinates in a sample genome that are not found to be adjacent in the reference genome. NanoVar characterizes SVs by featuring signals induced by novel adjacencies, and leverages the power of an artificial neural network to further improve the accuracy of SV detection.

Two tools were made available online in 2022. DeBreak, a hybrid method by integrating both alignment-based strategy and local assembly approach, separates SV events into two categories by whether the SV could be contained within reads. For SVs that could be spanned within reads, DeBreak extracts raw SV signals from read alignments and clusters them using a density-based clustering algorithm, and further refines the breakpoints with partial order alignment algorithm. For SV events that are too long to be contained in reads, DeBreak reconstructs the SV by performing local de novo assembly around candidate SV breakpoints which can be identified with abundant clipped alignment. MAMnet, another deep learning-based approach, first constructs feature vectors for each base pair that contains the count of substitution, deletion, insertion, soft/hard clip and read depth, as well as the max and average length of deletions and insertions that covers the base pair based on the alignment information, which in turn forms feature matrices for each subregion of the genome. These matrices are then processed sequentially with a time-distributed convolutional neural network (CNN) and a long short-term memory network (LSTM) to extract candidate deletion and insertion regions.

To date, only a few assembly-based tools have been proposed. Assembly-based tools detect SVs by parsing the alignments between the assembled contiguous sequences (contigs) and the reference genome. SVIM-asm, introduced in 2020, is adapted from SVIM and can accept both diploid and haploid assembly alignments as input. When running on diploid contig alignments, SVIM-asm extracts SV signatures for each haplotype and pairs similar signatures from different haplotypes for genotyping. PAV, which was proposed in 2021, first resolves multiply-mapped contig bases and reference bases by trimming alignment records, and then identifies SVs with precise breakpoints from inter- and intra-contig alignment. PAV also incorporates a special pipeline for inversion detection through a k-mer density analysis. Dipcall and Smartie-sv[19], both introduced in 2018, did not explicitly specify their SV calling pipelines in the respective papers introducing them, so their methods are not reviewed here, but we still investigate their performance and compare them with the rest of the SV callers.

## Aligners and assemblers for SV calling pipelines

To prepare the input BAM file for read alignment-based SV callers and further investigate the effect of different aligners on SV calling, three aligners were used to align each long-read dataset against the reference genome hg19: minimap2 (version 2.22-r1101), NGMLR (version 0.2.7), Winnowmap (version 2.03) and LRA (version 1.3.2). Minimap2 was run with the flag "-x" set to "map-hifi", "map-pb" and "map-ont" on PacBio-Hifi, PacBio-CLR, and ONT datasets, respectively. For NGMLR, the "-x" flag was set to "PacBio" or "ont" for PacBio or ONT datasets. Winnowmap, which was developed based on minimap2, used "map-pb", "map-pb-clr" and "map-ont" for the PacBio-Hifi, PacBio-CLR, and ONT datasets, respectively. LRA generated the alignment file by two steps: reference index and alignment. Both steps required a flag to specify the input type ("-CCS", "-CLR" and "-ONT" for PacBio-Hifi, PacBio-CLR, and ONT datasets, respectively). Additionally, minimap2 and Winnowmap were run with "−MD" flag, and LRA alignment step was run with "−printMD" to add an "MD" tag in alignment results while NGMLR included an "MD" tag in the output by default. The BAM file constitutes the input of most read alignment-based tools except SKSV. SKSV uses its own proposed pipeline to generate the alignment skeleton and extract SV signatures.

For assembly-based SV callers in this paper, a diploid/dual assembly is used as input. To prepare the diploid assembly for Hifi_L1, we adopted six de novo assemblers, Hifiasm (version 0.16)[43], HiCanu[44], Peregrine[45], Flye[46], IPA[47], and Wtdbg2[48]. To construct the diploid assembly for Nano_L1, three assemblers were used: Flye, Shasta[49], and Wtdbg2. For assemblers generating only haploid assembly, Peregrine, Flye, Wtdbg2, and Shasta, a dual assembly was subsequently generated using HapDup[50]; for assembler HiCanu, purge_dups[51] was used to generate a dual assembly.

Hifiasm is specially designed for PacBio Hifi datasets, and since version 0.15, Hifiasm uses a built-in dual assembly module to provide the assembly results of each haplotype. HiCanu generates a merged diploid contig file (e.g., 5.8Gb for Hifi_L1). Purge_dup is then used to split it into two haploid contig files which can be used for assembly-based SV callers such as Dipcall, SVIM-asm, and PAV. Peregrine outputs a primary contig (e.g. 2.9Gb for Hifi_L1) and an alternate contig fasta file (e.g. 318Mb for Hifi_L1) which can be directly used for SVIM-asm, and PAV. To prepare the diploid contigs of Peregrine for Dipcall, HapDup is further used to reconstruct the diploid information from the primary

contig. Flye and Wtdbg2 first assemble a collapsed haplotype contig file from the diploid reads data, and HapDup is then used to reconstruct the diploid information from it. IPA is designed for HiFi reads, and generates phased assemblies including a primary (e.g. 2.9Gb for Hifi_L1) and an alternate contig file (e.g., 2.5Gb for Hifi_L1), which can be directly used for Dipcall, SVIM-asm, and PAV. Shasta mainly focuses on assembling ONT data and offers a set of built-in assembly configurations that correspond to different guppy versions.

Assembly-based SV callers still require an alignment step in their workflow. It is either built-in (as in the case of PAV, Smartie-sv_asm, and Dipcall) or needs to be executed explicitly before running the SV caller (as in the case of SVIM-asm). In this paper, we used minimap2 to align assembled diploid contigs against the reference genome for all four of these assembly-based tools. Additional details for how to install and run each aligner and assembler are included on our GitHub page (https://github.com/maiziezhoulab/LRSV_combo). Details for how to install and run each SV caller are also included on our GitHub page. Several essential guidelines are included in the supplementary methods.

### Benchmarking for SV calls

The GIAB community provides a gold standard SV set for the sample HG002[52], which includes 4117 deletions and 5281 insertions in defined "high-confidence" regions characterized with multiple sequencing platforms. SV calls (deletions and insertions) from all tools were evaluated against this benchmark using Truvari, which is a commonly used open-source benchmarking tool.

Truvari provides parameters including pctstim ($p$), pctsize ($P$), pctovl ($O$), and refdist ($r$) to setup different criteria for SV evaluation depending on the needs of the specific analysis. Parameter $p$ controls the minimum allele sequence similarity used to identify two SV calls as identical. The similarity is calculated from the edit distance ratio between the reference and alternate haplotype sequences of the base and comparison call. Setting $p$ to zero can disable this comparison. Parameter $P$ corresponds to the minimum allele size similarity between the compared SVs, which is calculated from dividing the length of the shorter SV with the longer one. Parameter $O$ determines the minimum threshold of the reciprocal overlap ratio between the base and comparison call, and it is only applied to deletions for evaluating the effect of breakpoint shift on deletion accuracy. Parameter $r$ represents the threshold for maximum reference location difference of the compared SVs, which can be used to evaluate the effect of breakpoint shift on insertion accuracy. In general, higher values of $p$, $P$, and $O$, and lower value of $r$ set more stringent comparison criteria, as they will require the compared SVs to have higher sequence and size similarity, larger spatial overlapping ratio, or closer location to the reference sequence to be considered as the same SV.

### SV breakpoint shift and alternate allele sequence similarity analysis

Breakpoint shift is calculated from the reference genome location difference between the true positive SVs called by the tools and the corresponding benchmark SVs. Called SVs and benchmark SVs are paired up by the *MatchId* provided by Truvari. For each deletion, the start and end coordinate differences between the called SV and benchmark SV are calculated, and the maximum value of these two is chosen as the value for breakpoint shift. For insertions, the breakpoint shift is defined as the start coordinate difference. Breakpoint shift values larger than 200 bp are merged to the 200+bp bin in the distribution plot.

### SV overlapping analysis among different tools

To perform this overlapping analysis, a set of fixed and moderate-tolerance parameters ($p = 0$, $P = 0.5$, $r = 500$, $O = 0$) in Truvari was used. As mentioned before, parameter $p$ was set to zero to disable the SV

sequence comparison since five alignment-based tools do not provide alternate allele sequences for insertions, and the parameter O was set to zero to allow breakpoint shift for deletions, which was favorable for most alignment-based tools.

To determine the number of SV callers supporting each SV, we performed a SV overlapping analysis by employing Truvari to compare multiple VCFs of different SV callers. Truvari takes two VCFs as input, one as the SV benchmark set, and the other one as the SV comparison set. It then generates three VCF outputs to store true positive, false negative, and false positive results, respectively. The overlapping analysis requires SV calling results from different SV callers to have a compatible format. Therefore, VCF outputs from all SV callers were converted into a uniform format, with the corresponding tool name recorded in a user-defined field called "SC" (Source Caller) to allow us to backtrace all SV callers for each SV in the end. After this format regularization, we used Truvari to process these VCFs iteratively, as follows. The first SV caller's VCF file was treated as a merged VCF. Subsequent VCF files were compared against the merged VCF using Truvari to generate the true positive (tp-base.vcf), false negative (fn.vcf) and false positive (fp.vcf) results. As the merged VCF was used as the SV benchmark set in this comparison by Truvari, tp-base.vcf and fn.vcf shared the same SC information with the merged VCF, while fp.vcf shared the same SC field as the current VCF. Since SVs in tp-base.vcf were shared by the merged VCF and the current VCF, the tool name of the current VCF was then appended to the SC field of tp-base.vcf. SVs in fn.vcf and fp.vcf were either only found in the merged VCF or only in the VCF from the current tool, therefore SC information in these two VCFs remained unmodified. The modified tp-base.vcf, together with fn.vcf and fp.vcf were combined and used as the new merged VCF in the next iteration to compare with a third VCF of a different SV caller. Once all VCFs were used for comparisons, we generated a final merged VCF in which we kept track of all tools sharing each SV in the "SC" field. We thus used this merged VCF to calculate the number of tools supporting each SV.

### Orthogonal SV validation with a new complete human genome reference (T2T-CHM13) and trio-based Verkko HG002 assembly

To alleviate the bias introduced by the GIAB benchmark, we used a new complete human genome reference, T2T-CHM13, and designed a new pipeline to validate FP and FN SV calls inferred by each tool against the whole benchmark call set (N = 28745, without constraints). The FP and FN calls were validated by the same procedures and criteria described below, however, deletion and insertion SV calls were validated in a slightly different process.

For insertion SVs, in the first step, we inserted FP or FN SV calls (target SVs) into the reference genome. Each target SV was inserted into the hg19 reference genome by its estimated breakpoint recorded in the VCF file. Secondly, we simulated pseudo-reads containing all target SVs. For each insertion SV target, we simulated pseudo-reads from the SV-inserted hg19 reference genome in a way that each pseudo-read included a left and right flanking sequence length of 10 kb, plus the target SV sequence. So, the length of the corresponding pseudo-read is 20 kb + insertion SV size. For deletion SVs, instead of inserting each target SV, we deleted FP or FN SV calls from the hg19 reference genome by their breakpoints. We then simulated pseudo-reads which only contained two detached segments from the hg19 reference. So, the length of the corresponding pseudo-read is 20 kb. Thirdly, we used minimap2 to align all pseudo-reads to the T2T-CHM13 reference genome. The command we used was as follows:

```
minimap2 -t 30 --MD -Y -L -a -x map-
hifi T2T_genomic.fa pseudo-reads
.fa | samtools sort -o pseudo-
reads_aligned_T2T.bam
```

Finally, the last step was to filter SVs supported by T2T-CHM13. From the BAM file generated in the third step, we extracted the alignment information around SV breakpoints on pseudo-reads. For each insertion SV, we investigated the alignment information of the segment ranging from 9.9kb to 10.1 kb + SV size on the pseudo-read, as this segment encompasses the inserted SV and 100 bp left and right flanking regions. For each deletion SV, we investigated the alignment information of the segment from 9.9kb to 10.1kb, as this segment includes the novel adjacency caused by the target deletion. For the segment in question, we calculated the matching percentage $M$, deletion percentage $D$, and base shift $B$. The matching percentage is calculated as

$$M = N_{matching} / N_{seg} \times 100\% \tag{1}$$

where $N_{matching}$ is the number of matching bases in the segment aligned against T2T-CHM13, and $N_{seg}$ is the length of the segment. The deletion percentage is calculated as

$$D = \sum_{i=1}^{k} CDL_i / | \text{SV size} | \times 100\% \tag{2}$$

where $\sum_{i=1}^{k} CDL_i$ is the length summation of all deletion CIGARs in that segment region. The base shift is calculated as

$$B = |POS_{T2T} - POS_{hg19}| \tag{3}$$

$B$ is used as the absolute position difference between the breakpoint position on T2T-CHM13 and hg19. When the segment was aligned to a different chromosome on T2T-CHM13 compared to hg19, B was set to ∞. For a target SV, it was counted as supported by T2T-CHM13 when the below condition holds:

$$M > 95\% \quad \text{and} \quad D < 5\% \quad \text{and} \quad B < 10Mb \tag{4}$$

We also used the trio-based Verkko HG002 assembly for orthogonal SV validation. The evaluation pipeline is the same as using T2T, except it does not need to check the chromosome number and base shift since Verkko assembly is not chromosome-level assembly.

## Subsampling

In order to evaluate the consequences of different sequencing coverage on the performance of the SV callers, we used a tool called rasusa[39] to subsample the raw reads of Hifi_L1 to approximately 50×, 40×, 30×, 20×, 10× and 5x coverage, and of Nano_L1 to approximately 40×, 30×, 20×, 10× and 5x coverage. The original coverage for Hifi_L1 and Nano_L1 was approximately 56.3x and 45.6x fold, respectively. Rasusa takes the read length into account while subsampling the dataset to a certain coverage and thus generates unbiased random subsamples of long-read data. These subsamples were then aligned or assembled to serve as the input to corresponding SV calling pipelines.

## Simulation of complex SVs in different types of long reads datasets

To investigate complex SVs such as translocations, inversions, and duplications, we simulated 9 PacBio Hifi, CLR, and ONT datasets with known gold standards (Table 1). In our simulations, 380 reciprocal translocations and 3712 duplications from KWS1 sample callsets (nstd106 in dbVAR database), along with 44 inversions from CHM1 sample callsets (nstd137 in dbVAR database) were included. We first generated diploid genomes. These three types of SVs were selected based on the same criteria as in Jiang et al.[22]. They were inserted separately into the human reference genome (hg19) with VISOR[53], generating three in silico genomes (Hap1_TRA, Hap1_DUP, and

Hap1_INV). To simulate genotypes, we constructed haplotype 2 (Hap2) for each SV type (Hap2_TRA, Hap2_DUP and Hap2_INV), where chromosomes were randomly selected to be homozygous or heterozygous to mimic homozygous and heterozygous SVs. For homozygous chromosomes, Hap2 was a copy of Hap1, whereas, in heterozygous chromosomes, Hap2 was identical to the reference genome. These simulated diploid genomes were then fed into PBSIM3[54] to generate simulated Pacbio and ONT long reads.

We simulated 40× coverage reads with PBSIM3 for each diploid genome, where half of the reads were from Hap1 and half were from Hap2. ONT and Pacbio CLR reads were simulated with PBSIM3 ERRHMM-ONT model and ERRHMM-SEQUEL model respectively. For Pacbio HiFi reads, we first simulated multi-pass Pacbio CLR reads with PBSIM3 ERRHMM-SEQUEL model and then used ccs software[55] to generate HiFi reads. Additional details for how to install and simulate reads are included in our GitHub page.

## Evaluation of translocation, inversions, and duplications in simulated datasets

In order to compare the performance of different tools in detecting translocations (TRA), inversions (INV) and duplications (DUP), we ran each tool on simulated datasets and evaluated the performance against the benchmark callset relying on the following procedure.

For TRAs, we compared each tool's callset and benchmark at the breakend level. As we described in the Methods section, the benchmark callset includes 380 reciprocal translocations, equaling 1520 breakends, as every reciprocal translocation includes 4 breakends. Every breakend can be written as a signature

$$(chrom1, pos1, strand1, chrom2, pos2, strand2)$$

where chrom1, pos1, chrom2, pos2 are the breakpoints of two chromosome segments that are attached together during a TRA event, and the strand is the direction of each chromosome segment, which could be either + (forward strand) or − (reverse strand). We extracted all breakends from the VCF file of each tool, transferred them into signatures, and compared them against the benchmark using the below five criteria. Each breakend in the callset was considered TP if it satisfied the following criteria with one record in the benchmark set; otherwise, it was considered FP:

$$\begin{cases} chrom1_{call} = chrom1_{bench} \\ chrom2_{call} = chrom2_{bench} \\ |pos1_{call} - pos1_{bench}| \leq 1kb \\ |pos2_{call} - pos2_{bench}| \leq 1kb \\ strand1_{call} = strand1_{bench} \\ strand2_{call} = strand2_{bench} \end{cases}$$

For INVs and DUPs, the evaluation criteria are similar to TRA. Every inversion or duplication can be written as a signature in the form of

$$(chrom, start, end, SVsize, GT)$$

which includes chromosome number, SV start position, SV end position, SV size, and genotype information, respectively. The evaluation criteria are as below

$$\begin{cases} chrom_{call} = chrom_{bench} \\ |start_{call} - start_{bench}| \leq 500bp \\ |end_{call} - end_{bench}| \leq 500bp \\ P \geq 0.5 \end{cases}$$

where $P$ is the size similarity between the call and benchmark and is calculated by the formula that follows

$$P = \min(\text{SVsize}_{\text{call}}, \text{SVsize}_{\text{bench}}) / \max(\text{SVsize}_{\text{call}}, \text{SVsize}_{\text{bench}}) \quad (5)$$

If an INV or DUP meets the criteria above with one record in the benchmark set, it is considered TP, otherwise FP. Furthermore, if an INV/DUP meets the criteria above, and has a matching GT with one record in the bench set, it is considered TP_GT.

### SV discovery in cancer genomes

To compare the ability of different SV calling methods in identifying somatic mutations, we used the high-confidence HCC1395 somatic SV callset and the Pacbio and ONT Tumor-Normal paired libraries of HCC1395 from Talsania et al.[42] in our analysis (Table 1). We first applied each SV caller on two sets of Tumor-Normal paired libraries, and then adopted the SV filtering and merging tool SURVIVOR[56] to extract somatic mutations. Lastly, we compared the somatic SV call set generated by each SV caller to the high-confidence somatic SV call set.

The pipeline details were described as follows. The first step is to split VCF file by SV type and SV size window. We first split each VCF file into a TRA (translocation) VCF file and a non-TRA VCF file, as SURVIVOR has a different algorithm tailored to TRA merging than to other types of SV merging. We further split the non-TRA VCF file into 5 size windows: 50–100 bp, 101–500 bp, 501–1000 bp, 1001–30,000 bp, >30,000 bp. The command used was

```
SURVIVOR filter ${input_vcf} NA
${min_sv_size}${max_sv_size} 0 -1
${out_vcf}
```

Secondly, we extracted somatic mutations by merging the tumor and normal SV call set. For a VCF file of a certain size window, we used the lower bound of its size window as the maximum threshold of breakpoint distance when merging. For example, for a 50-100 bp VCF file, the breakpoint distance threshold was set to 50 bp. The command used was

```
SURVIVOR merge ${vcflist.txt}
${dist_thresh} 1 1 0 0
${min_sv_size} ${merged.vcf}
```

where vcflist.txt includes one VCF file for the normal SV set and another VCF file for the tumor SV set. After merging, we filtered SVs that were only supported by the tumor call set but not by the normal call set as somatic SVs. We performed this filtering procedure on non-TRA VCF files of different size windows and the TRA VCF file and concatenated six filtered VCF files into one VCF file as the final somatic mutation call set.

The last step was to evaluate somatic mutation against the high-confidence call set using different criteria. We evaluated TRAs only on the breakend level. We calculated the breakend shift $B$ between TRA from the call set and the benchmark set and if $B \le 1kb$, this TRA was considered TP, otherwise FP. For non-TRA SVs, we calculated both the breakpoint shift $B$ and size similarity $P$. $P$ is calculated as

$$P = \min(\text{SVsize}_1, \text{SVsize}_2) / \max(\text{SVsize}_1, \text{SVsize}_2) \quad (6)$$

if $B \le 500$ bp and $P \ge 0.5$, this non-TRA SV was considered TP, otherwise FP.

### Reporting summary

Further information on research design is available in the Nature Portfolio Reporting Summary linked to this article.

## Data availability

PacBio CLR, HiFi, and ONT HG002 sequencing reads are available at GIAB and NCBI. The high-confidence HCC1395 somatic SV callset and the Pacbio and ONT Tumor-Normal paired libraries of HCC1395 are publicly accessible at NCBI. Specific links for all 15 aforementioned real datasets are listed in Table 1. The Tier1 benchmark SV callset and high-confidence HG002 region were obtained from https://ftp-trace.ncbi.nlm.nih.gov/ReferenceSamples/giab/data/AshkenazimTrio/analysis/NIST_SVs_Integration_v0.6/. T2T assembly and sequencing reads of CHM13 are publicly available at https://github.com/marbl/CHM13. Trio-based Verkko HG002 assembly was obtained from https://zenodo.org/record/7400747/files/hg002_verkko_hifi_ont_trio.fasta.gz. SV callsets for each method evaluated in the paper are deposited at https://zenodo.org/record/8287836. Source data are provided with this paper.

## Code availability

All customized benchmarking scripts and detailed information for installing and running each investigated method are available at https://github.com/maiziezhoulab/LRSV_combo under the MIT License[57].

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

## Acknowledgements

This work was supported by the NIH NIGMS Maximizing Investigators' Research Award (MIRA) R35 GM146960 to X.M.Z.

## Author contributions

X.M.Z. conceived and led this work. Y.H.L and C.L. collected all datasets and testing of all the methods. Y.H.L and C.L. implemented all customized benchmarking pipelines. Y.H.L, C.L, S.G.G, and J.B.I. performed data analysis. Y.H.L, C.L, and X.M.Z. wrote the manuscript.

## Competing interests

The authors declare no competing interests.
