## [Peer Review File · Nature Communications]

Tradeoffs in alignment and assembly-based methods for structural variant detection with long-read sequencing dataReviewer #1 (Remarks to the Author):

This manuscript by Liu et al provides a comparative analysis of structural variant detection algorithms on long-read sequencing data. They compared 12 read alignment-based SV callers, and 4 assembly-based SV callers, along with 4 upstream aligners and 7 assemblers in the study, to assess the relative merits of different tools.

The presentation of the results itself is well developed. The author considered many scenarios including down-sampling and the evaluation of effects of different aligners, which should be applauded. The figures are also generally quite comprehensive in their comparative analysis. I have a few comments below that I feel the author should address to make the paper more useful and informative to the community. Additionally, I feel that the paper does not actually present a concrete conclusion from their extensive comparative analysis, so as a reader, there is not specific take home message that I can see clearly.

Major comments.

1, The reliance on GIAB SV gold standard callset is a major concern. I am not saying that it is not reliable, but that a single data set is not representative enough to showcase the accuracy of different software tools, especially when many software tools actually used this exact same data to fine-tune parameters. Additionally, even if they want to stick with one single samples, I do not think they actually used all available data (they only used HiFi L1, CLR L1, CLR L2, CLR L3, and Nano L1.); the pangenome consortium itself has a lot more data https://github.com/human-pangenomics/HG002_Data_Freeze_v1.0 including ultralong Nanopore reads which should be a lot more informative for SV detection. A typical reader will wonder how ultralong reads can change the landscape of SV analysis on human genomes. PacBio also released HiFi data with improved chemistry on HG002 which should be more accurate than Illumina and a typical reader will wonder how this will change the way SV can be identified with single base resolution. Further, PacBio has varying library insert sizes (from 10kb to 30 kb) in their HiFi protocol and it is interesting to see how this impact SV detection, and all such data has been released to the public domain on HG002.

2, Continue with the above statement, as geneticists, we always feel parental information should be used in judgment of computational software tools that analyze genetic mutations. Mendelian consistency is among the best measure of accuracy of variant calling including SV calling. The long-read data, including Nanopore and PacBio, are indeed available for parents of HG002 (I believe they are commonly referred to as HG003 and HG004). The authors should perform analysis (does not have to be as comprehensive as HG002) on parental data and provides orthogonal evaluation metrics on the performance of long-read SV callers in terms of Mendelian consistency. This avoided any potential biases introduced by GIAB gold standard set.

3, The specific focus on insertions vs. deletions is a bit too narrow, and is more or less a result of the lack of benchmarking data. In practice, structural variants include translocations and inversions and complex rearrangements due to a combination of events, and it can be more challenging than inferring copy number changes alone. Some software tools used simulation data to evaluate the performance of SV calling, and I think the authors can do the same thing but in a more independent manner to assess how different methods have different results. There are many methods that can generate simulation data on long-read sequencing nowadays, but it is also not difficult to write one; there are quite a few databases documenting inversions and translocations (many of which are located in complex genomic regions, or are the results of microhomology) that can be used to generate simulation data.

4, Additionally, there are a number of widely used cancer cell lines for which translocations (including gene fusions) are well known and reported, and I believe that long-read data is available on either the PacBio or Nanopore platform or both, and these cancer lines can be a good real benchmarking data on

structural variants other than copy number changes.

5, While Truvari is indeed a widely used tool for evaluation, it has its own limitations such as the ability to handle haplotype resolved SV calls. One example is hap-eval (<https://github.com/Sentieon/hap-eval>) which is used for evaluation of SV call sets. It is an open source tool under active development. For SV callers that allows haplotype-resolved calls (especially assembly based method), it would be ideal to use this algorithm to evaluate the performance since the results can be more informative to understand SVs.

Minor comments:

1, the title is a bit misleading "Methods for structural variant detection with long-read sequencing data". Perhaps they can rephrase it to indicate what specific goals (comparative evaluation, etc) are presented in the manuscript.

2, The author used "copy number polymorphisms (CNPs)" term in the paper. Generally speaking, whenever polymorphism is used, it indicates a relatively common or recurrent copy number changes. I think what they want to say is copy number variants (CNVs) which include polymorphic changes as well as rare or private changes.

3, The authors claimed that Sequencing accuracy approaches 99% for HiFi reads. This is not wrong, but a context needs to be given that such accuracy is a result of consensus calling, and it critically depends on the actual coverage (how many sub-reads are sequenced to generate the consensus call). It can be much higher than 99% at least as claimed by the company. Using something like Q20/Q30 scores under a specific coverage is probably more appropriate to describe this technology.

4, The statement about "PacBio and ONT are both hindered by lower throughput, higher error rate, and higher cost per base" can be challenged by some readers. As mentioned earlier, the accuracy for PacBio HiFi is already a lot higher than Illumina. For PacBio, R9 cells does not have high error rate any more, and R10 cells have accuracy comparable o Illumina. The cost per base for ONT is also fairly comparable to Illumina at the moment under optimal sequencing conditions. I also do not think it is fair to say lower throughput, because both Illumina and ONT have different sequencers with different scales of sequencing capacity/throughput.

5, It is better to say HG002 (NA24385) so that readers understand that genomes are used in the benchmarking study. Many readers are only familiar with the term HG002.

6, the abstract does not really include any conclusion to be informative to readers. Something like "assembly-based tools are superior in detecting large insertions in long-read datasets with sufficient coverage (>50x)" would be more informative to be included in an abstract of a paper.

7, Fonts for some of the figures need to be changed, since they are too small (Figure 2, 3, etc). Figure 3 probably do not need the actual numbers in the cells, since it is impossible to see the numbers; color scale of the cells would be good enough.

8, Information for each SV caller is provided in Table S1. Since this table is not a large table, it may be better presented in the main text, since the version number and the software name is important for readers to know immediately. Some software tools can yield very different results depending on the specific version.

Reviewer #2 (Remarks to the Author):

The authors provide a review article on long read SV calling, where they briefly survey existing tools and apply them to existing Genome-in-a-Bottle benchmarking data sets. I list my comments and concerns below.

Major comments

- One important metric that many readers would be able to relate to is the total number of SVs detected per sample. Right now, these numbers are not readily visible and I found myself summing up different statistics shown. Stratifying by size (e.g. in Figure 2) is good, but for a main figure it would also be helpful to show the grand total. That would also reveal that (apparently) all callers seem to call relatively few SVs. The total number of SVs per sample has been shown to be between 20k and 30k, varying a bit depending on alignment method used and on ancestry of the sample. This study mostly pertains to only a relatively small subset of ~9k SVs represented in the GIAB benchmark. As a general point, much more focus should be to the calls outside the GIAB benchmark. So in my view creating better benchmark data sets is much more important for the community than running many tools on existing and known-to-be incomplete benchmarks.
- The part of the paper aiming to address this (Figure 5) is confusing. The total number of calls (FPs+TPs) seem to sum up to much smaller numbers compared to what's visible in Figure 2. Also, especially the assembly-based tools should find many more SVs in total.
- Using a benchmark based on hg19 is problematic. Even GRCh38 is known to contain issues and gaps in regions prone to SV, which is partly addressed by the CHM13-T2T genome reference.
- An evaluation of genotyping performance is missing. That is, you should break the two questions apart about whether a tool is able to detect a variant at all and whether it is able to assign the correct genotype.
- The title of the paper is misleading as there are no new methods introduced in this paper. The title should reflect that this is a study comparing different tools.
- The introduction provides quick summaries of what the different tools do. I'm not convinced that this is helpful in this form as it is not self-contained. That is, if you really want to survey SV calling techniques, then you need to expand on the methodological ideas and possibly create something like a feature matrix showing which techniques are used by which of the tools.
- Analyzing/discussing caller performance by different SV classes (in term of there mutational mechanisms), would be helpful.
- Be consistent about which set of tools is used: i) Sniffles2, DeBreak and MAMnet are mentioned and included in the overview on Figure 1, but then they're not included in most of the analysis. ii) Abstract says 12+4 SV callers, introduction says 15 callers (fig. 1 has 15, with Smartie-sv being on both sides), and results says 16 callers. Please, be consistent on the number of callers.

Minor comments

- The paper lacks a evaluation/discussion on how the examined tools perform for further variant types, such as duplications, inversions and potentially more complex types of variants (or why that wasn't

analyzed).

- I think Figure 6 has lots of information that can be reduced using a better presentation for a main display item, while some of these details could go to a supplement. One option would be precision-recall curves both for HiFi and ONT data on the same plot.

- You use NanoSV to run both ONT and PacBio data. However, this is not a fair comparison because this tool intends to use Nanopore data. I would suggest removing NanoSV runs from PacBio data analysis or discussing this accordingly.

- The promise of the last sentence of the abstract "Based on these results, we offer guidelines and recommendations for different tools and directions for further method development." is not really met in my view, because it is not clear to me what insights for method development have been made in this paper.

- Introduction: "SVs are larger alterations that span more than 50 base pairs and they also involve insertions or deletions, in addition to copy number polymorphisms (CNPs)". This is misleading as insertions and deletions are copy number variants.

- "However, PacBio and ONT are both hindered by lower throughput, higher error rate, and higher cost per base" The error rate of HiFi/CCS is comparable to Illumina today. For this sentence, an outdated citation from 2015 does not make much sense.

- "... and compare the assembly with the reference genome...", I suggest to replace this with "... with a reference genome...".

- "Information for each SV caller is provided in Table S1" Which information? Please be more specific.

- "... we performed multiple SV calling and evaluation experiments on subsampled datasets in a later section." In which section?

- Please add information about the configuration of your machine for the observed runtimes.

- Please provide a citation for the "Rasusa" algorithm in "Subsampling effects of SV calling"

- "To perform this analysis, a set of fixed parameters ($p=0$, $P=0.5$, $r=500$) was used." Please indicate that this pertains to "Truvari" here.

- It's fine that your command lines for each tool is in your Github repo, but please also include them in the Supplement.

- I didn't understand why you divide the figures into two in Figure 2? Please merge a-b; c-d; e-f; and g-h in the same plot. Also in the same figure, CPU time consumptions are black and white, whereas the others are colored.

- The size of the headers, titles of the axes and legends of your figures don't seem to be consistent. Some are very small (e.g., Figure 3, Figure S2), but some look fine. Please make them more readable and consistent.

- Smartie-sv is in both categories (alignment-based and assembly-based). This should be made explicit and justified in the introduction

- In the overview section, sniffles2 should be together with the other signature-based methods (just like cuteSV and SKSV are), or, at least, presented right after them.

- Figure 2a-h is a bit confusing: it's not explicit where each legend applies, and same colors are used for different tools
- "SV callers often consumed less than 100Gb memory, except for PAV, Smartie-sv, DeBreak, and NanoSV (Table S3)." Table S3 only shows values above 100Gb for memory in Smartie-sv for Nano_L1 and PAV for Nano_L1. The four enumerated tools show values higher than 100 but for CPU hours.
- "all assemblers consumed 200-400Gb memory (Table S4)" IPA and wtdbg2 didn't, at least for the Hifi_L1 dataset
- "Although F1 scores in these tools increased significantly to the range of 70.1% - 85.5% after disabling the sequence comparison" The lowest value of the referred tools is 68.7%, by NanoVar.
- On the evaluation of SV calls, 4th paragraph, it's worth mentioning that Smartie-sv's F1, in the assembly-based methods, for Hifi_L1, goes from 9.4% to 91.2% when changing the value of p.
- p, P and r Truvari parameters are mentioned in "Evaluation of SV calls among existing alignment and assembly-based tools" but only explained after in "Evaluation of SV calls across parameters"
- "and even maintained a greater than 70% F1 score" pbsv is 69.9%, it's not greater than 70%
- Fig.3a,b,e,f values are hard to read
- "With respect to deletion detection using the most relaxed parameters ($p = 0$ and $O = 0$) (Figure S21a), NanoSV, Sniffles, SVIM, and cuteSV achieved excellent F1 scores" NanoSV is by far the worst, maybe you meant NanoVar
- "The other two tools, Smartie-sv, and NanoVar displayed the opposite trend, and their F1 scores dropped faster on Nano L1 than on Hifi L1" Smartie-sv doesn't show that pattern, as the solid lines all drop below the dashed lines for high values of O.
- Fig.S21: lines hard to distinguish, and colors for $p=0.0$ and $p=1.0$ are very similar
- Fig.S22: missing proper identification of deletions and insertions.
- Table S11: how come that Merge2 has 4,049 TP when PAV alone has 4,050 TP?
- Fig.S23: why wasn't this done for assembly-based callers?
- Fig.S24 is not mentioned appropriately in the results section.
- "Sniffles, SVIM, pbsv, cuteSV, and SKSV, achieved good performance under the fixed and optimal evaluation parameters setting, they were very sensitive to choice of parameters. In contrast, assembly-based approaches such as Dipcall, SVIM-asm, and PAV were more robust" pbsv, as mentioned in the Results section, showed a pattern more similar to the assembly-based methods, and so shouldn't be clustered with the other alignment-based callers here.
- "Alignment-based tools, such as cuteSV, are less affected by different aligners compared to other tools." Sentence is not clear, it looks like all alignment-based tools are less affected by different aligners, but then there wouldn't be other tools to compare. Maybe you meant that "some alignment-based tools" are less affected than others.
- LRA aligner is not mentioned in the "Methods" section.

- Why didn't you include PacBio's standard aligner pbmm2?

Typos

- Intro: "resourcess"
- "To reveal the underlying reason read alignment based..." Missing comma
- "These results indicate that most of TP SV calls are supported by most of tools" should be "... most of the tools"
- Nanopore instead of "The NanoPore dataset..."
- "Sniffles": the tool name is misspelled multiple times
- "We also investigated the subsampling effects on SV calling on Nano L1 (Figure S24)" Figure S23, not S24.
- "using assembly-based SV callers with subsampled Hifi L1 data (Figure 6g-i)" Figure 6g-l, instead of 6g-i

We appreciated the expert review of our article titled “Methods for structural variant detection with long-read sequencing data”. We have made extensive changes to the manuscript to address all issues raised. We believe the revised version is substantially strengthened as a result.

Reviewer #1

This manuscript by Liu et al provides a comparative analysis of structural variant detection algorithms on long-read sequencing data. They compared 12 read alignment-based SV callers, and 4 assembly-based SV callers, along with 4 upstream aligners and 7 assemblers in the study, to assess the relative merits of different tools.

The presentation of the results itself is well developed. The author considered many scenarios including down-sampling and the evaluation of effects of different aligners, which should be applauded. The figures are also generally quite comprehensive in their comparative analysis. I have a few comments below that I feel the author should address to make the paper more useful and informative to the community. Additionally, I feel that the paper does not actually present a concrete conclusion from their extensive comparative analysis, so as a reader, there is not specific take home message that I can see clearly.

0. I feel that the paper does not actually present a concrete conclusion

Response: We realize that we did not make our conclusion clear enough in the original manuscript and we have worked to rectify that in the revised draft. We revised the Discussion section and added Table 2, to make our results more informative to readers. We generated 2-4 key conclusions for each results section and highlighted the optimal tools based on 28 criteria. We also suggested future method development directions.

Major comments.

1. The reliance on GIAB SV gold standard callset is a major concern. I am not saying that it is not reliable, but that a single data set is not representative enough to showcase the accuracy of different software tools, especially when many software tools actually used this exact same data to fine-tune parameters. Additionally, even if they want to stick with one single samples, I do not think they actually used all available data (they

only used HiFi L1, CLR L1, CLR L2, CLR L3, and Nano L1.); the pangenome consortium itself has a lot more data https://github.com/human-pangenomics/HG002_Data_Freeze_v1.0 including ultralong Nanopore reads which should be a lot more informative for SV detection. A typical reader will wonder how ultralong reads can change the landscape of SV analysis on human genomes. PacBio also released HiFi data with improved chemistry on HG002 which should be more accurate than Illumina and a typical reader will wonder how this will change the way SV can be identified with single base resolution. Further, PacBio has varying library insert sizes (from 10kb to 30 kb) in their HiFi protocol and it is interesting to see how this impact SV detection, and all such data has been released to the public domain on HG002.

Response: The reviewer raises a valid point. In the original manuscript, we used a sampling of libraries to illustrate our points, however, we do see the benefit of further expanding our datasets and we now expand our analysis to the datasets mentioned and several additional ones. These include 2 more recent ONT and 4 more PacBio HiFi datasets, plus 2 sets of HCC1395 Tumor-Normal paired libraries (Pacbio and ONT) and 9 simulated long reads datasets (Table 1). For these new datasets, we added two more results sections: “SV calling performance across 11 different PacBio and ONT datasets” and “SV calling performance on cancer datasets”.

2. Continue with the above statement, as geneticists, we always feel parental information should be used in judgment of computational software tools that analyze genetic mutations. Mendelian consistency is among the best measure of accuracy of variant calling including SV calling. The long-read data, including Nanopore and PacBio, are indeed available for parents of HG002 (I believe they are commonly referred to as HG003 and HG004). The authors should perform analysis (does not have to be as comprehensive as HG002) on parental data and provides orthogonal evaluation metrics on the performance of long-read SV callers in terms of Mendelian consistency. This avoided any potential biases introduced by GIAB gold standard set.

Response: The reviewer raises another important point about the use of parental data, however, we respectfully submit that this is not an issue for the GIAB gold standard set. We did not use parental information to further evaluate SV sets because the GIAB gold standard set (Zook et al, Nature Biotechnology 2020) already used trio data to filter and refine the SV gold standard set of HG002. We quote the corresponding text and table from their paper:

- “The 30,062 SVs remaining were then evaluated and genotyped in each member of the trio using svviz to align reads to reference and alternate alleles from PCR-free

Illumina, Illumina 6-kbp mate-pair, haplotype-partitioned 10× Genomics and Pacific Biosciences with and without haplotype partitioning.”

- *“We further filtered for SVs covered in HG002 by eight or more Pacific Biosciences reads (mean coverage of about 60), with at least 25% of Pacific Biosciences reads supporting the alternate allele and consistent genotypes from all technologies that could be confidently assessed with svviz.”*
- *“For SVs on autosomes, we also identified if genotypes were consistent with Mendelian inheritance. When limiting to 7,973 autosomal SVs in the benchmark set for which a consensus genotype from svviz was determined for both of the parents, only 20 violated Mendelian inheritance.”*

RESOURCE

NATURE BIOTECHNOLOGY

Father		0/0			0/1			1/1		
Mother		0/0	0/1	1/1	0/0	0/1	1/1	0/0	0/1	1/1
Son	0/1	14	1185	417	1143	1119	462	416	522	12
	1/1	0	0	0	0	449	444	2	431	2748

Extended Data Fig. 2 | Mendelian contingency table for sites with consensus genotypes from svviz in the son, father, and mother. SVs in boxes highlighted in red violate the expected Mendelian inheritance pattern. Variants on chromosomes X and Y are excluded.

However, to expand our orthogonal evaluation, we further extended our overlapping SV analysis to all GIAB benchmark SVs (without constraints), and designed a new pipeline by employing a new complete human genome reference, T2T-CHM13, to validate all false positive and false negatives (Figure 4). The corresponding results and methods were described in results section “Orthogonal SV validation with overlapping calls among different tools and a new complete human genome reference (T2T-CHM13)” and methods section “Orthogonal SV validation with a new complete human genome reference (T2T-CHM13)”, respectively.

3. *The specific focus on insertions vs. deletions is a bit too narrow, and is more or less a result of the lack of benchmarking data. In practice, structural variants include translocations and inversions and complex rearrangements due to a combination of events, and it can be more challenging than inferring copy number changes alone. Some software tools used simulation data to evaluate the performance of SV calling, and I think the authors can do the same thing but in a more independent manner to assess how different methods have different results. There are many methods that can generate simulation data on long-read sequencing nowadays, but it is also not difficult*

to write one; there are quite a few databases documenting inversions and translocations (many of which are located in complex genomic regions, or are the results of microhomology) that can be used to generate simulation data.

Response: The reviewer's point is well taken. Large deletions and insertions account for the most abundant SVs, but other SVs such as translocations and inversions also describe different combinations of DNA rearrangements. We simulated 9 long reads datasets to benchmark translocations, inversions and duplications (Table 1). The corresponding results and methods were described in results section "Evaluation of complex SVs with simulated datasets" and methods sections "Simulation of complex SVs in different types of long read datasets" and "Evaluation of translocation, inversion and duplications in simulated datasets", respectively.

4. Additionally, there are a number of widely used cancer cell lines for which translocations (including gene fusions) are well known and reported, and I believe that long-read data is available on either the PacBio or Nanopore platform or both, and these cancer lines can be a good real benchmarking data on structural variants other than copy number changes.

Response: We do see the reviewer's point and we now performed additional analysis on two publicly available sets of Tumor-Normal paired libraries (sequenced with PacBio and ONT). The corresponding results and methods are described in the results section "SV calling performance on cancer datasets" and methods section "SV discovery in cancer genomes", respectively.

5. While Truvari is indeed a widely used tool for evaluation, it has its own limitations such as the ability to handle haplotype resolved SV calls. One example is hap-eval (<https://github.com/Sentieon/hap-eval>) which is used for evaluation of SV call sets. It is an open source tool under active development. For SV callers that allows haplotype-resolved calls (especially assembly based method), it would be ideal to use this algorithm to evaluate the performance since the results can be more informative to understand SVs.

Response: The reviewer's point is well taken. We used "hap-eval" tool to evaluate SV calls for most tools in FigureS18-S20. We discussed the results in the results section titled "Evaluation of SV calls across parameters".

Minor comments:

1. the title is a bit misleading “Methods for structural variant detection with long-read sequencing data”. Perhaps they can rephrase it to indicate what specific goals (comparative evaluation, etc) are presented in the manuscript.

Response: We appreciate the reviewer’s point. We changed our paper title to “A speed-performance tradeoff in alignment and assembly-based methods for structural variant detection with long-read sequencing data”.

2. The author used “copy number polymorphisms (CNPs)” term in the paper. Generally speaking, whenever polymorphism is used, it indicates a relatively common or recurrent copy number changes. I think what they want to say is copy number variants (CNVs) which include polymorphic changes as well as rare or private changes.

Response: The reviewer’s point is well taken. We fixed this in the paper.

3. The authors claimed that Sequencing accuracy approaches 99% for HiFi reads. This is not wrong, but a context needs to be given that such accuracy is a result of consensus calling, and it critically depends on the actual coverage (how many sub-reads are sequenced to generate the consensus call). It can be much higher than 99% at least as claimed by the company. Using something like Q20/Q30 scores under a specific coverage is probably more appropriate to describe this technology.

Response: The reviewer’s point is well taken. We revised this part and cited a new paper.

4. The statement about “PacBio and ONT are both hindered by lower throughput, higher error rate, and higher cost per base” can be challenged by some readers. As mentioned earlier, the accuracy for PacBio HiFi is already a lot higher than Illumina. For PacBio, R9 cells does not have high error rate any more, and R10 cells have accuracy comparable o Illumina. The cost per base for ONT is also fairly comparable to Illumina at the moment under optimal sequencing conditions. I also do not think it is fair to say lower throughput, because both Illumina and ONT have different sequencers with different scales of sequencing capacity/throughput.

Response: The reviewer’s point is well taken. We revised this part and cited a new paper.

5. *It is better to say HG002 (NA24385) so that readers understand that genomes are used in the benchmarking study. Many readers are only familiar with the term HG002.*

Response: The reviewer's point is well taken. We used HG002 instead of NA24385 in the paper.

6. *the abstract does not really include any conclusion to be informative to readers. Something like "assembly-based tools are superior in detecting large insertions in long-read datasets with sufficient coverage (>50x)" would be more informative to be included in an abstract of a paper.*

Response: We agreed with the reviewer's point that it is good to include informative conclusions for readers. Therefore, we generated 2-4 key conclusions for each results section and highlighted the optimal tools based on 28 criteria. We now provide a general overview of these results in the abstract. We have also revised the Discussion section and added Table 2, to make our results more informative to readers.

7. *Fonts for some of the figures need to be changed, since they are too small (Figure 2, 3, etc). Figure 3 probably do not need the actual numbers in the cells, since it is impossible to see the numbers; color scale of the cells would be good enough.*

Response: The reviewer's point is well taken. We fixed these in Figure 2 and 3.

8. *Information for each SV caller is provided in Table S1. Since this table is not a large table, it may be better presented in the main text, since the version number and the software name is important for readers to know immediately. Some software tools can yield very different results depending on the specific version.*

Response: The reviewer's point is well taken. We remade Table 1.

Reviewer #2

The authors provide a review article on long read SV calling, where they briefly survey existing tools and apply them to existing Genome-in-a-Bottle benchmarking data sets. I list my comments and concerns below.

Major comments

1. One important metric that many readers would be able to relate to is the total number of SVs detected per sample. Right now, these numbers are not readily visible and I found myself summing up different statistics shown. Stratifying by size (e.g. in Figure 2) is good, but for a main figure it would also be helpful to show the grand total. That would also reveal that (apparently) all callers seem to call relatively few SVs. The total number of SVs per sample has been shown to be between 20k and 30k, varying a bit depending on alignment method used and on ancestry of the sample. This study mostly pertains to only a relatively small subset of ~9k SVs represented in the GIAB benchmark. As a general point, much more focus should be to the calls outside the GIAB benchmark. So in my view creating better benchmark data sets is much more important for the community than running many tools on existing and known-to-be incomplete benchmarks.

Response: The reviewer raises an important point, and we include the total number of SVs for each tool in table S3. It is not accurate, however, that we called relatively few SVs, or fewer than the expected 20-30k number of SVs in the genome. For example, we called 25-26k SVs with two popular SV calling tools (cuteSV and PAV) in Hifi_L1, exactly at the expected range. We believe the confusion came about from Figure 2, which only plotted large autosomic deletions and insertions. We now present full data in Table S3, a part of which is reproduced below:

tool	>=50 bp (all SVs)	>=50bp (autosome, INS & DEL) Total SV calls in Figure 2
cuteSV (alignment-based)	25910	18023
PAV (assembly-based)	26242	23218

It is also not correct that the GIAB gold standard SV set (Zook et al, Nature Biotechnology 2020) is incomplete. The dataset involves over **28k** SVs. We focused on

a subset of ~9k SVs from the golden standard because these were the most high-confidence ones. These SVs were validated and filtered by several robust strategies and were the fairest ones for benchmarking. We will now also expand our analysis to the full set of 28k gold standard SVs for our benchmarking in Figure 4.

2. The part of the paper aiming to address this (Figure 5) is confusing. The total number of calls (FPs+TPs) seem to sum up to much smaller numbers compared to what's visible in Figure 2. Also, especially the assembly-based tools should find many more SVs in total.

Response: This criticism stems from the confusion about the benchmark SVs we used in the original Figure 5 (now part of Figure 4). For this analysis, we only show True Positive and False Positive SV calls validated based on the ~9k high-confidence SVs identified in GIAB. To clarify this point, we added evaluation results in Figure 4 based on all benchmark SVs (N = 28745, without constraints). The total SVs we plotted in Figure 2 were 18k for cuteSV, and 23k for PAV (shown in Table 1 above). Assembly-based tools like PAV did find more SVs than alignment-based tools like cuteSV, especially for large insertions, which is consistent with Figure 2 and Figure 4. One of our key conclusions was also that “assembly-based tools detected more large insertions than most alignment-based tools, especially greater than 1kb”.

3. Using a benchmark based on hg19 is problematic. Even GRCh38 is known to contain issues and gaps in regions prone to SV, which is partly addressed by the CHM13-T2T genome reference.

Response: The hg19 benchmark (on which the gold-standard is based) is the best available set, however, we appreciate the reviewer's point for the CHM13-T2T genome reference, and we have expanded our orthogonal evaluation. We further extended our overlapping SV analysis to all GIAB benchmark SVs (without constraints) and designed a new pipeline by employing a new complete human genome reference, T2T-CHM13, to validate all false positive and false negatives (Figure 4). The corresponding results and methods are described in the results section “Orthogonal SV validation with overlapping calls among different tools and a new complete human genome reference (T2T-CHM13)” and methods section “Orthogonal SV validation with a new complete human genome reference (T2T-CHM13)”, respectively.

4. *An evaluation of genotyping performance is missing. That is, you should break the two questions apart about whether a tool is able to detect a variant at all and whether it is able to assign the correct genotype.*

Response: The reviewer's point is well taken. We in fact had results of genotyping performance in hand but we elected not to include them in the original paper, for simplicity. We now add these results in the revised version in Figures 5, 6, S20, and Table S3-8.

5. *The title of the paper is misleading as there are no new methods introduced in this paper. The title should reflect that this is a study comparing different tools.*

Response: In response to both reviewers, we have revised our paper title to "A speed-performance tradeoff in alignment and assembly-based methods for structural variant detection with long-read sequencing data".

6. *The introduction provides quick summaries of what the different tools do. I'm not convinced that this is helpful in this form as it is not self-contained. That is, if you really want to survey SV calling techniques, then you need to expand on the methodological ideas and possibly create something like a feature matrix showing which techniques are used by which of the tools.*

Response: We appreciate the reviewer's suggestion. We remade Table 1 to show detailed information for all tools and datasets. As the approach of each tool is quite unique rather than relying on a combination of standard features, we provide explanation of features and methods in the narrative.

7. *Analyzing/discussing caller performance by different SV classes (in term of there mutational mechanisms), would be helpful.*

Response: Reviewer 1, question #3 raised the same issue. The reviewers' point is well taken. Large deletions and insertions account for the most abundant SVs, but other SVs such as translocations and inversions also describe different combinations of DNA rearrangements. The lack of benchmarking data makes it difficult to evaluate tools. We simulated 9 long reads datasets to benchmark three types of complex SV classes: translocations, inversions and duplications (Table 1). The corresponding results and methods are described in the results section "Evaluation of complex SVs with simulated datasets" and methods sections "Simulation of complex SVs in different types of long

readd datasets” and “Evaluation of translocation, inversion and duplications in simulated datasets”, respectively.

We also performed additional analysis on two publicly available sets of Tumor-Normal paired libraries (Pacbio and ONT) to investigate three complex SV classes and deletions and insertions. The corresponding results and methods were described in results section “SV calling performance on cancer datasets” and methods section “SV discovery in cancer genomes”, respectively.

8. Be consistent about which set of tools is used: i) Sniffles2, DeBreak and MAMnet are mentioned and included in the overview on Figure 1, but then they're not included in most of the analysis. ii) Abstract says 12+4 SV callers, introduction says 15 callers (fig. 1 has 15, with Smartie-sv being on both sides), and results says 16 callers. Please, be consistent on the number of callers.

Response: The reviewer’s point is well taken. i) We included these three tools in the late stage of our analysis (two of them are still in preprint) for the purposes of completeness, and included their results in supplemental figures 10, 11 and 24. Since they did not demonstrate significantly better results than other representative alignment-based tools such as cuteSV and pbsv, we had not included them in most of the main figures.

We have now updated all our main figures, tables and supplementary information to make them more consistent. We also kept each tool’s color consistent in all figures.

ii) Because Smartie-sv has both “alignment-based” and “assembly-based” modes, we now denote it as two methods: “Smartie-sv_aln” and “Smartie-sv_asm” in the paper text, figures and tables. We also clarified these numbers.

Minor comments

1. The paper lacks a evaluation/discussion on how the examined tools perform for further variant types, such as duplications, inversions and potentially more complex types of variants (or why that wasn't analyzed).

Response: The reviewer’s point is well taken. The lack of benchmarking data made it difficult to evaluate complex SVs. For this reason, we simulated 9 long reads datasets to benchmark three types of complex SV classes: translocations, inversions and duplications (Table 1). The corresponding results and methods are described in the results section “Evaluation of complex SVs with simulated datasets” and methods

sections “Simulation of complex SVs in different types of long read datasets” and “Evaluation of translocation, inversion and duplications in simulated datasets”, respectively.

We also performed additional analysis on publicly available two sets of Tumor-Normal paired libraries (Pacbio and ONT) to investigate three complex SV classes and deletions and insertions. The corresponding results and methods were described in results section “SV calling performance on cancer datasets” and methods section “SV discovery in cancer genomes”, respectively.

2. I think Figure 6 has lots of information that can be reduced using a better presentation for a main display item, while some of these details could go to a supplement. One option would be precision-recall curves both for HiFi and ONT data on the same plot.

Response: We appreciate the reviewer’s point. We now use precision-recall-F1 curves in this figure (now figure 5).

3. You use NanoSV to run both ONT and PacBio data. However, this is not a fair comparison because this tool intends to use Nanopore data. I would suggest removing NanoSV runs from PacBio data analysis or discussing this accordingly.

Response: We used NanoSV to run PacBio data because the authors of NanoSV pointed out NanoSV could be used for PacBio data in GitHub. Quoting the authors: “NanoSV is a software package that can be used to identify structural genomic variations in long-read sequencing data, such as data produced by Oxford Nanopore Technologies’ MinION, GridION or PromethION instruments, or Pacific Biosciences RSII or Sequel sequencers.”

4. The promise of the last sentence of the abstract “Based on these results, we offer guidelines and recommendations for different tools and directions for further method development.” is not really met in my view, because it is not clear to me what insights for method development have been made in this paper.

Response: We appreciate the reviewer’s point. We revised the Discussion section for this purpose and provided more detailed tables and results.

5. *Introduction: “SVs are larger alterations that span more than 50 base pairs and they also involve insertions or deletions, in addition to copy number polymorphisms (CNPs)”. This is misleading as insertions and deletions are copy number variants.*

Response: We appreciate the reviewer’s point. We rephrased it.

6. *“However, PacBio and ONT are both hindered by lower throughput, higher error rate, and higher cost per base” The error rate of Hifi/CCS is comparable to Illumina today. For this sentence, an outdated citation from 2015 does not make much sense.*

Response: The reviewer’s point is well taken. We revised this part and cited a new paper.

7. *“... and compare the assembly with the reference genome...”, I suggest to replace this with “... with a reference genome...”.*

Response: The reviewer’s point is well taken. We fixed this.

8. *“Information for each SV caller is provided in Table S1” Which information? Please be more specific.*

Response: The reviewer’s point is well taken. Table S1 was moved to be Table 1 in the revised version. We add details for this information in Table 1.

9. *“... we performed multiple SV calling and evaluation experiments on subsampled datasets in a later section.” In which section?*

Response: The reviewer’s point is well taken. We fixed this.

10. *Please add information about the configuration of your machine for the observed runtimes.*

Response: The reviewer’s point is well taken. We added the configuration of our nodes in the legend of Table S1 and Table S2.

11. *Please provide a citation for the “Rasusa” algorithm in “Subsampling effects of SV calling”.*

Response: The reviewer’s point is well taken. We added this citation.

12. *“To perform this analysis, a set of fixed parameters ($p=0$, $P=0.5$, $r=500$) was used.” Please indicate that this pertains to “Truvari” here.*

Response: The reviewer’s point is well taken. We fixed this.

13. *It’s fine that your command lines for each tool is in your Github repo, but please also include them in the Supplement.*

Response: The reviewer’s point is well taken. We added command lines in the Supplementary information.

14. *I didn’t understand why you divide the figures into two in Figure 2? Please merge a-b; c-d; e-f; and g-h in the same plot. Also in the same figure, CPU time consumptions are black and white, whereas the others are colored.*

Response: We divided these figures into two because large ($\geq 1\text{kb}$) SVs only account for a small percentage of total SVs. Breaking the figure into two was necessary to visualize the large SV distribution when using a small y-axis range. We also wanted to differentiate SVs by alignment-based tools versus assembly-based tools by using two figures. However, we now use a box to highlight all these figures together.

15. *The size of the headers, titles of the axes and legends of your figures don’t seem to be consistent. Some are very small (e.g., Figure 3, Figure S2), but some look fine. Please make them more readable and consistent.*

Response: The reviewer’s point is well taken. We corrected these in all figures.

16. *Smartie-sv is in both categories (alignment-based and assembly-based). This should be made explicit and justified in the introduction*

Response: The reviewer’s point is well taken. We clarified this in the paper text, figures, and tables.

17. *In the overview section, sniffles2 should be together with the other signature-based methods (just like cuteSV and SKSV are), or, at least, presented right after them.*

Response: The reviewer’s point is well taken. We fixed this in the overview section.

18. *Figure 2a-h is a bit confusing: it's not explicit where each legend applies, and same colors are used for different tools*

Response: The reviewer's point is well taken. We revised the legend and kept all tools' color consistent in all figures.

19. *"SV callers often consumed less than 100Gb memory, except for PAV, Smartie-sv, DeBreak, and NanoSV (Table S3)." Table S3 only shows values above 100Gb for memory in Smartie-sv for Nano_L1 and PAV for Nano_L1. The four enumerated tools show values higher than 100 but for CPU hours.*

Response: Thanks for this point. We corrected this statement in the paper.

20. *"all assemblers consumed 200-400Gb memory (Table S4)" IPA and wtdbg2 didn't, at least for the Hifi_L1 dataset*

Response: Thanks for this point. We corrected this statement in the paper.

21. *"Although F1 scores in these tools increased significantly to the range of 70.1% - 85.5% after disabling the sequence comparison" The lowest value of the referred tools is 68.7%, by NanoVar.*

Response: In view of this point, we have decided to remove this section from the paper, for clarity and to focus on the main message of the results.

22. *On the evaluation of SV calls, 4th paragraph, it's worth mentioning that Smartie-sv's F1, in the assembly-based methods, for Hifi_L1, goes from 9.4% to 91.2% when changing the value of p.*

Response: In view of this point, we have decided to remove this section from the paper, for clarity and to focus on the main message of the results. In the result section titled "Evaluation of SV calls across parameters", we do highlight the change of F1 as a result of the change of p .

23. *p , P and r Truvari parameters are mentioned in "Evaluation of SV calls among existing alignment and assembly-based tools" but only explained after in "Evaluation of SV calls across parameters"*

Response: Thanks for this point. We added more explanation for these parameters before the section “*Evaluation of SV calls across parameters*”.

24. *"and even maintained a greater than 70% F1 score" pbsv is 69.9%, it's not greater than 70%*

Response: Thanks for this point. We corrected this percentage to “greater than 69%”.

25. *Fig.3a,b,e,f values are hard to read*

Response: Thanks for this point. We adjusted the font in the figure.

26. *"With respect to deletion detection using the most relaxed parameters ($p = 0$ and $O = 0$) (Figure S21a), NanoSV, Sniffles, SVIM, and cuteSV achieved excellent F1 scores" NanoSV is by far the worst, maybe you meant NanoVar*

Response: In view of this point, we have decided to remove this section from the paper, for clarity and to focus on the main message of the results.

27. *"The other two tools, Smartie-sv, and NanoVar displayed the opposite trend, and their F1 scores dropped faster on Nano L1 than on Hifi L1" Smartie-sv doesn't show that pattern, as the solid lines all drop below the dashed lines for high values of O.*

Response: same previous comment 26, in view of this point, we have decided to remove this section from the paper, for clarity and to focus on the main message of the results.

28. *Fig.S21: lines hard to distinguish, and colors for $p=0.0$ and $p=1.0$ are very similar*

Response: same previous comment 26, in view of this point, we have decided to remove this section from the paper, for clarity and to focus on the main message of the results.

29. *Fig.S22: missing proper identification of deletions and insertions.*

Response: Thanks for this point. We remade this figure (now Figure S19) and added the proper identification.

30. *Table S11: how come that Merge2 has 4,049 TP when PAV alone has 4,050 TP?*

Response: Thanks for this point. We used Truvari for merging results. This tool may lose TPs due to its imperfect merging process in some rare scenarios. We deleted this table and revised results for this part. We only kept the Merge3 results (merge SVs from alignment-based and assembly-based tools) and reported all numbers in the paper.

31. *Fig.S23: why wasn't this done for assembly-based callers?*

Response: Thanks for this point. We added this analysis in Figure S19c-d.

32. *Fig.S24 is not mentioned appropriately in the results section.*

Response: Based on the reviewer's major comment #8, we updated all figures and tables for "MAMnet, DeBreak and Sniffles2", and revised the corresponding text.

33. *"Sniffles, SVIM, pbsv, cuteSV, and SKSV, achieved good performance under the fixed and optimal evaluation parameters setting, they were very sensitive to choice of parameters. In contrast, assembly-based approaches such as Dipcall, SVIM-asm, and PAV were more robust" pbsv, as mentioned in the Results section, showed a pattern more similar to the assembly-based methods, and so shouldn't be clustered with the other alignment-based callers here.*

Response: Thanks for this point. We corrected this statement.

34. *"Alignment-based tools, such as cuteSV, are less affected by different aligners compared to other tools." Sentence is not clear, it looks like all alignment-based tools are less affected by different aligners, but then there wouldn't be other tools to compare. Maybe you meant that "some alignment-based tools" are less affected than others.*

Response: Thanks for this point. We corrected this statement.

35. *LRA aligner is not mentioned in the "Methods" section.*

Response: Thanks for this point. We added the description for LRA in the methods section.

36. *Why didn't you include PacBio's standard aligner pbmm2?*

Response: Thanks for this point. We only selected four aligners which can align both PacBio and ONT datasets for comparisons.

Typos

- Intro: "resourcess"
- "To reveal the underlying reason read alignment based..." Missing comma
- "These results indicate that most of TP SV calls are supported by most of tools" should be "... most of the tools"
- Nanopore instead of "The NanoPore dataset..."
- "Sniffiles": the tool name is misspelled multiple times
- "We also investigated the subsampling effects on SV calling on Nano L1 (Figure S24)" Figure S23, not S24.
- "using assembly-based SV callers with subsampled Hifi L1 data (Figure 6g-i)" Figure 6g-l, instead of 6g-i

Response: We wish to thank the reviewer. We have corrected these typos, rewritten some of these sentences, and proof-read the manuscript carefully.

Reviewer #1 (Remarks to the Author):

This is a substantially revised manuscript to discuss alignment and assembly-based methods for structural variants (SV) detection from long-read sequencing. The authors examined 12 read alignment-based SV calling methods, and 4 assembly-based SV calling methods, along with 4 upstream aligners and 7 assemblers. With these changes mentioned in the response letter and a change of title, I think the message is more clear compared to the previous version of the manuscript. Most of the responses to my previous comments are addressed well. I have some additional comments or new comments below.

A practical issue is the dichotomization of alignment and assembly-based methods: there are also reference guided assembly approaches that do not require as much computational resource (or even sequence coverage) as de novo whole-genome assembly. Many alignment-based algorithms nowadays implement some flavors of local assembly after alignments are done; this is not just the case for SV calling, but also for SNP calling and indel calling as well. I think they could soften the dichotomization a bit to acknowledge that hybrid approaches are available (additionally Figure 1 has a very clear dichotomization that can be perhaps modified to show that these two schools of thoughts are not mutually exclusive). Also see the comments below about machine learning approaches.

Several machine-learning based approaches are published in recent years, including some ones this year. This concept is quite distinct from alignment and assembly based methods. For example, some methods take alignment file, but build specific feature sets from the alignment, and some methods directly convert the alignment profiles of each read against the reference genome into an image that is fed to a convolutional neural network. Examples are SVision, Cue, MaMNet and some others. If it is not too much additional work (not my intention to ask for more work but these methods are all pretty new), I do feel that the authors can probably evaluate these newly published methods, using the existing evaluation framework that they have already established so that there is a head-to-head comparison with older generation of computational methods. I think as readers, we will be curious to see how this new generation of computational methods compare with previous approaches especially since the current publications all seem to suggest that deep learning approaches (built on alignment) work much better than traditional methods that evaluate alignment itself.

While I appreciate the effort to make Table 2, partly to address the comments on lack of clear message to readers, I also think this table is overly complex. As an end user of software tools (rather than a developer), I still cannot really tell exactly what is the message to the community. Say for example, for "capable of detecting large deletion" and "capable of detecting large duplication", the vast majority of methods have a check in the box; but this is more to tell readers that the existing computational methods can do these routine (really the most basic type) SV calling, not really a recommendation of what to use, as there are still a dozen or more methods that have a check mark. (personally, I would rather see something like "top 2 best" rather than a list of over one dozen tools, otherwise I would still have to evaluate a dozen tools myself before deciding which tool to use to call large deletions or duplications which defeats the purpose of this manuscript.) The 28 criteria also make the table really complicated to interpret and I do really think a regular reader or software user can extract specific guidance from this table: this table should show what the authors feel (from their analysis in this paper), not what the developers advertise, so that it can offer some insights to readers of the manuscript.

In Figure 6, it is not really feasible to see the differences since all these colored lines overlap with each other (perhaps because there is not much differences between the methods). Perhaps a different presentation of the same set of data/results would be helpful, since in the current form, I cannot really see a difference (especially in Nano, compared to HiFi, in panel b).

Figure 7, somewhere in the paper, the authors should explain what "HiFi gt", "CLR gt" mean. I think they meant genotyping, but this abbreviation is never explained anywhere in the paper or in figure

legend. The F1 scores are also surprisingly low, for example, for Sniffles, only 0.00054 for CLR and only 0.049 for HiFi and 0.01 for ONT. This is quite concerning: either the methods do perform extremely poorly, or that the authors have simulations that do not fit reality. Since Sniffles is perhaps among the most widely used SV callers, I think some additional investigation is needed to better explain the results. I understand that "The lack of benchmarking data makes it difficult to evaluate tools", and I think it is reasonable to use simulations, but the results are just a bit too concerning. Also, in the main manuscript, they did not explicitly mention this, but instead say "maximum genotype F1 drop" which somehow hides this issue of very poor performance.

A minor issue: The first sentence in abstract ends with "in principle." which reads a bit strange. I think the sentence can be re-organized.

Reviewer #2 (Remarks to the Author):

The authors added a lot of analyses to the paper, which provide additional information. Still, I do not agree with the authors that "The hg19 benchmark (on which the gold-standard is based) is the best available set". Even if the unconstrained set is larger, it is based on fairly old data. In my view, leveraging modern genome assemblies (e.g. produced from a combination of Hifi and ONT-UL reads (e.g. through tools like Verkko or hybrid Hifi-Asm) would lead to much informative analyses. CHM13 is added as a confirmation for SV alleles shared between HG002 and CHM13. That is a helpful analysis, but it would be more informative to compare to an assembly of the same sample. Further points:

- Overall, the paper would need to be streamlined. Both text and figures are too dense. I would recommend to focus the readers attention to the important aspects that lead to concrete conclusions and recommendations and defer exhaustive analyses to the supplement.

- An analysis on genotyping has been added to the figures, but is not really discussed/interpreted in the text.

- The revised title "A speed-performance tradeoff in alignment and assembly-based methods for structural variant detection with long-read sequencing data" suggest that one specific trade-off is recommended in the paper, which is not the case.

- "all three PacBio sequencing platforms (RS II, Sequel, and Sequel II)": That is outdated, there is Revio and Onso now.

Response to Reviewers

We appreciate the reviewers' careful reading of our manuscript and have performed additional work to address all of the remaining comments.

Reviewer #1:

This is a substantially revised manuscript to discuss alignment and assembly-based methods for structural variants (SV) detection from long-read sequencing. The authors examined 12 read alignment-based SV calling methods, and 4 assembly-based SV calling methods, along with 4 upstream aligners and 7 assemblers. With these changes mentioned in the response letter and a change of title, I think the message is more clear compared to the previous version of the manuscript. Most of the responses to my previous comments are addressed well. I have some additional comments or new comments below.

1. A practical issue is the dichotomization of alignment and assembly-based methods: there are also reference guided assembly approaches that do not require as much computational resource (or even sequence coverage) as de novo whole-genome assembly. Many alignment-based algorithms nowadays implement some flavors of local assembly after alignments are done; this is not just the case for SV calling, but also for SNP calling and indel calling as well. I think they could soften the dichotomization a bit to acknowledge that hybrid approaches are available (additionally Figure 1 has a very clear dichotomization that can be perhaps modified to show that these two schools of thoughts are not mutually exclusive). Also see the comments below about machine learning approaches.

Response: The reviewer's point is well taken. We now revised the corresponding text parts for most of the sections to soften the dichotomization to acknowledge both the hybrid and deep learning-based approaches. We also modified Figure 1 for this reason.

2. Several machine-learning based approaches are published in recent years, including some ones this year. This concept is quite distinct from alignment and assembly based methods. For example, some methods take alignment file, but build specific feature sets from the alignment, and some methods directly convert the alignment profiles of each read against the reference genome into an image that is fed to a convolutional neural network. Examples are SVision, Cue, MAMNet and some others. If it is not too much additional work (not my intention to ask for more work but these methods are all pretty new), I do feel that the authors can probably evaluate these newly published methods, using the existing evaluation framework that they have already established so that there is a head-to-head comparison with older generation of computational methods. I think as readers, we will be curious to see how this new generation of computational methods compare with previous approaches especially since the current publications all seem to suggest that deep learning approaches (built on alignment) work much better than traditional methods that evaluate alignment itself.

Response: We see the reviewer's point. In final result section and the supplementary results section 2.6, we additionally benchmarked the two most recent deep learning-based SV calling methods (SVision and INSnet) using our established evaluation framework and discussed their strength and weakness compared to traditional alignment-based methods. More discussion for deep learning-based and hybrid methods is also included in the discussion section.

MAMnet was already included in the previous version of the manuscript, but Cue was not included in our benchmark framework since they did not provide guidelines or a trained model for SV detection with long reads in their paper and GitHub. We recently asked again the authors for this, and they replied "Planning to release fully trained models for PacBio sometime by the end of the summer" (In response to our request, these authors provide details about this lack of capability in their Github page: <https://github.com/PopicLab/cue/issues/14>).

3. While I appreciate the effort to make Table 2, partly to address the comments on lack of clear message to readers, I also think this table is overly complex. As an end user of software tools (rather than a developer), I still cannot really tell exactly what is the message to the community. Say for example, for "capable of detecting large deletion" and "capable of detecting large duplication", the vast majority of methods have a check in the box; but this is more to tell readers that the existing computational methods can do these routine (really the most basic type) SV calling, not really a recommendation of what to use, as there are still a dozen or more methods that have a check mark. (personally, I would rather see something like "top 2 best" rather than a list of over one dozen tools, otherwise I would still have to evaluate a dozen tools myself before deciding which tool to use to call large deletions or duplications which defeats the purpose of this manuscript.) The 28 criteria also make the table really complicated to interpret and I do really think a regular reader or software user can extract specific guidance from this table: this table should show what the authors feel (from their analysis in this paper), not what the developers advertise, so that it can offer some insights to readers of the manuscript.

Response: We appreciated the reviewer's suggestion. We redesigned table 2 to highlight tools by ranking. Even though it includes 31 combinations of criteria for both PacBio and ONT data, it is pretty straightforward to select optimal tools based on users' needs. This table tracks the results section and makes it easier for readers to follow the evaluation methods and benchmarking results.

4. In Figure 6, it is not really feasible to see the differences since all these colored lines overlap with each other (perhaps because there is not much differences between the methods). Perhaps a different presentation of the same set of data/results would be helpful, since in the current form, I cannot really see a difference (especially in Nano, compared to HiFi, in panel b).

Response: The reviewer's point is well taken. We now adjusted the axis of Figure 6 to make the difference more visible.

5. Figure 7, somewhere in the paper, the authors should explain what "HiFi gt", "CLR gt" mean. I think they meant genotyping, but this abbreviation is never explained anywhere in the paper or in figure legend. The F1 scores are also surprisingly low, for example, for Sniffles, only 0.00054 for CLR and only 0.049 for HiFi and 0.01 for ONT. This is quite concerning: either the methods do perform extremely poorly, or that the authors have simulations that do not fit reality. Since Sniffles is perhaps among the most widely used SV callers, I think some additional investigation is needed to better explain the results. I understand that "The lack of benchmarking data makes it difficult to evaluate tools", and I think it is reasonable to use simulations, but the results are just a bit too concerning. Also, in the main manuscript, they did not explicitly mention this, but instead say "maximum genotype F1 drop" which somehow hides this issue of very poor performance.

Response: The reviewer makes a good point. We clarified the "gt" abbreviation in both text and figure legend. We investigated Sniffles's extremely low genotype accuracy by double-checking its genotype errors. Sniffles did assign most of its complex SVs with the wrong genotype and had a bias in the direction of calling homozygous SVs as heterozygous ones. Its low genotype accuracy was also reflected in both deletion and insertion SVs (Figure 5e, blue dashed line). Their new version Sniffles2 did improve the genotype accuracy of all types of SVs significantly.

We also did some literature review for Sniffle's GT problem. cuteSV paper also observed that sniffles reached a much lower GT F1 for all types of SVs than other tools (Figure 2 and 3 in reference 20).

6. A minor issue: The first sentence in abstract ends with "in principle." which reads a bit strange. I think the sentence can be re-organized.

Response: We revised this sentence.

Reviewer #2:

1. The authors added a lot of analyses to the paper, which provide additional information. Still, I do not agree with the authors that "The hg19 benchmark (on which the gold-standard is based) is the best available set". Even if the unconstrained set is larger, it is based on fairly old data. In my view, leveraging modern genome assemblies (e.g. produced from a combination of Hifi and ONT-UL reads (e.g. through tools like Verkko or hybrid Hifi-Asm) would lead to much informative analyses. CHM13 is added

as a confirmation for SV alleles shared between HG002 and CHM13. That is a helpful analysis, but it would be more informative to compare to an assembly of the same sample. Further points:

Response: The reviewer's point is well taken. We add trio-based Verkko HG002 assembly with T2T for orthogonal SV validation at our results section. Verkko assembly and T2T gave us highly concordant evaluation results when benchmarking all tools.

2. Overall, the paper would need to be streamlined. Both text and figures are too dense. I would recommend to focus the readers attention to the important aspects that lead to concrete conclusions and recommendations and defer exhaustive analyses to the supplement.

Response: We appreciate the reviewer's suggestion. We moved some analyses from the main results section to the supplementary results sections 2.1-2.2 and 2.4-2.9. We also moved the "Overview of existing SV callers" section to Methods section. Based on Reviewer 1's comment, we also redesigned the recommendation table 2 and made it easier for readers to follow the evaluation methods, our comprehensive benchmarking results, and our conclusions.

3. An analysis on genotyping has been added to the figures, but is not really discussed/interpreted in the text.

Response: The reviewer's point is well taken. We now highlight and add more analysis for the genotyping results.

One issue we also want to point out here is that the genotyping results for three assembly-based tools (PAV, SVIM-asm, and Dipcall) that we reported in our previous version were not entirely correct for some conditions. The reason was an error in the benchmarking tool Truvari. We corrected this and have reported this in this tool's GitHub page.

4. The revised title "A speed-performance tradeoff in alignment and assembly-based methods for structural variant detection with long-read sequencing data" suggest that one specific trade-off is recommended in the paper, which is not the case.

Response: The reviewer's point is well taken. We have revised our paper title to "Tradeoffs in alignment and assembly-based methods for structural variant detection with long-read sequencing data".

5. *"all three PacBio sequencing platforms (RS II, Sequel, and Sequel II)": That is outdated, there is Revio and Onso now.*

Response: The reviewer's point is well taken. We revised the related introduction part.

Reviewer #1 (Remarks to the Author):

The authors did a good job responding to my comments and I think the paper is substantially improved. Congratulations.

One cosmetic issue is with Tables, for example in Table 2, where three different colors are used to represent different types of methods. Since this information could be provided elsewhere, for example an extra column in Table 1, I think it is better not to use colored cells in these tables.

Also Table 1, the authors may want to explicitly mark (a) and (b) for the two sub-tables. Right now they all look like the same table even though the column headers are different in two sub-tables.

Response to Reviewers

We appreciate the reviewers' careful reading of our manuscript and have addressed the remaining minor comments.

Reviewer #1:

The authors did a good job responding to my comments and I think the paper is substantially improved. Congratulations.

One cosmetic issue is with Tables, for example in Table 2, where three different colors are used to represent different types of methods. Since this information could be provided else where, for example an extra column in Table 1, I think it is better not to use colored cells in these tables.

Response: Thanks for your positive feedback overall! We revised both tables.